# Transcriptomic and epigenetic regulation of hair cell regeneration in the mouse utricle and its potentiation by Atoh1

Hsin-I Jen[1], Matthew C Hill[1], Litao Tao[2,3], Kuanwei Sheng[4], Wenjian Cao[5], Hongyuan Zhang[6], Haoze V Yu[2,3], Juan Llamas[2,3], Chenghang Zong[5], James F Martin[1,7,8], Neil Segil[2,3], Andrew K Groves[1,6]*

[1]Program in Developmental Biology, Baylor College of Medicine, Houston, United States; [2]Department of Stem Cell Biology and Regenerative Medicine, Keck School of Medicine, University of Southern California, Los Angeles, United States; [3]Caruso Department of Otolaryngology - Head and Neck Surgery, Keck School of Medicine, University of Southern California, Los Angeles, United States; [4]Program in Integrative Molecular and Biomedical Sciences, Baylor College of Medicine, Houston, United States; [5]Department of Molecular and Human Genetics, Baylor College of Medicine, Houston, United States; [6]Department of Neuroscience, Baylor College of Medicine, Houston, United States; [7]Department of Molecular Physiology and Biophysics, Baylor College of Medicine, Houston, United States; [8]The Texas Heart Institute, Houston, United States

**Abstract** The mammalian cochlea loses its ability to regenerate new hair cells prior to the onset of hearing. In contrast, the adult vestibular system can produce new hair cells in response to damage, or by reprogramming of supporting cells with the hair cell transcription factor Atoh1. We used RNA-seq and ATAC-seq to probe the transcriptional and epigenetic responses of utricle supporting cells to damage and Atoh1 transduction. We show that the regenerative response of the utricle correlates with a more accessible chromatin structure in utricle supporting cells compared to their cochlear counterparts. We also provide evidence that Atoh1 transduction of supporting cells is able to promote increased transcriptional accessibility of some hair cell genes. Our study offers a possible explanation for regenerative differences between sensory organs of the inner ear, but shows that additional factors to Atoh1 may be required for optimal reprogramming of hair cell fate.

DOI: https://doi.org/10.7554/eLife.44328.001

*For correspondence: akgroves@bcm.edu

**Competing interests:** The authors declare that no competing interests exist.

## Introduction

Sensory hair cells are exquisitely sensitive mechanosensors present in the inner ear and lateral line organs of vertebrates. They are extremely vulnerable to the mechanical trauma of environmental noise exposure, and to ototoxic aminoglycoside antibiotics and platinum-containing chemotherapeutics (*Oesterle, 2013*; *Liberman, 2016*; *Francis and Cunningham, 2017*; *Jiang et al., 2017*; *Liberman, 2017*; *Sheth et al., 2017*). Non-mammalian vertebrates are able to regenerate significant numbers of hair cells and achieve impressive functional recovery following deafening (*Brignull et al., 2009*; *Ryals et al., 2013*; *Monroe et al., 2015*), and naturally replenish hair cells in their vestibular and lateral line sensory organs by an ongoing process of self-renewal (*Corwin, 1981*; *Corwin, 1985*; *Jørgensen and Mathiesen, 1988*; *Roberson et al., 1992*; *Kil et al., 1997*). These regenerative processes involve the mobilization of neighboring supporting cells to re-enter the cell cycle and trans-

differentiate into hair cells (*Cotanche, 1987*; *Corwin and Cotanche, 1988*; *Ryals and Rubel, 1988*; *Stone and Cotanche, 2007*). In contrast, the mammalian hearing organ, the organ of Corti, has an extremely limited capacity for spontaneous regeneration, and this has only been observed in immature or embryonic mammals (*Kelley et al., 1995*; *Bramhall et al., 2014*; *Cox et al., 2014*; *Atkinson et al., 2015*). This regenerative failure has prompted attempts to promote mammalian hair cell regeneration experimentally. A variety of manipulations, performed mostly in immature mice and rats, can induce neonatal supporting cells to divide and trans-differentiate into hair cell-like cells (*Atkinson et al., 2015*). These include placing supporting cells in dissociated cell culture (*White et al., 2006*; *Oshima et al., 2007*; *Sinkkonen et al., 2011*), inhibition of the Notch signaling pathway, and activation of the canonical Wnt signaling pathway (*Shi et al., 2013*; *Atkinson et al., 2015*; *Kuo et al., 2015*; *Maass et al., 2015*; *Zak et al., 2015*; *Hu et al., 2016*; *Maass et al., 2016*; *Ni et al., 2016*). In addition, forced expression of the hair cell-specific transcription factor Atoh1 can cause supporting cells and non-sensory cells of the cochlea to convert to hair cell-like cells (*Zheng and Gao, 2000*; *Kawamoto et al., 2003*; *Izumikawa et al., 2005*; *Zhao et al., 2011*; *Kelly et al., 2012*; *Yang et al., 2012a*; *Yang et al., 2012b*; *Chen et al., 2013*; *Atkinson et al., 2014*; *Kuo et al., 2015*; *Lee et al., 2017*). However, the mammalian cochlea becomes refractory to these various manipulations as the organ of Corti matures, and almost no regeneration can be induced after the onset of hearing (*Kelly et al., 2012*; *Liu et al., 2012*). As supporting cells are neither added nor replaced in the organ of Corti during this period, it is likely that maturation of cochlear supporting cells is responsible for their failure to divide and differentiate into hair cells. However, the mechanistic basis for this maturation remains elusive.

In contrast to the mammalian cochlea, mature mammalian vestibular organs are capable of a very limited, but significant, amount of ongoing hair cell production and regeneration following damage (*Forge et al., 1993*; *Rubel et al., 1995*; *Kuntz and Oesterle, 1998*; *Ogata et al., 1999*; *Kawamoto et al., 2009*; *Lin et al., 2011*; *Golub et al., 2012*). Small numbers of immature hair cells have been observed in the adult mammalian utricle, including in humans (*Taylor et al., 2015*), and recent use of lineage tracing techniques have clearly demonstrated that small numbers of supporting cells can trans-differentiate into cells resembling type II hair cells in the adult mouse utricle, at a rate of about two hair cells added per week (*Bucks et al., 2017*). This rate of replacement increases significantly after hair cells are experimentally killed (*Bucks et al., 2017*). Hair cell production in the adult utricle is superficially similar to that observed during embryonic hair cell formation in mammals: it involves up-regulation of the Atoh1 transcription factor, and is regulated by the Notch signaling pathway (*Wang et al., 2010*; *Lin et al., 2011*; *Golub et al., 2012*; *Bucks et al., 2017*). Indeed, over-expression of the Atoh1 transcription factor in normal or ototoxin-treated adult utricle organ cultures or immature mice leads to a robust production of hair cell-like cells (*Shou et al., 2003*; *Gao et al., 2016*; *Taylor et al., 2018*).

These results raise a number of questions regarding the capacity of mature supporting cells to contribute to hair cell regeneration and the capacity of Atoh1 to enhance this regeneration. First, what is the transcriptional response of supporting cells to hair cell loss, and are any aspects of the hair cell differentiation program activated in these cells in the absence of overt hair cell production? Second, are the hair cell-like cells produced in the utricle spontaneously after damage, or after transduction of Atoh1, *bona fide* hair cells, as opposed to a supporting cell-hair cell hybrid? Third, why are mature utricle supporting cells apparently more competent to trans-differentiate into hair cells than their cochlear counterparts? In the present study, we have addressed these questions using a utricle organ culture model of hair cell damage, combined with RNA-seq and ATAC-seq analysis of supporting cells. We find that hair cell loss alone leads to up-regulation of many characteristic hair cell genes in supporting cells, although these cells do not express typical hair cell markers such as Myosin7a. Transduction of these cultures with an Atoh1-expressing adenovirus induces significant numbers of Myosin7a-expressing hair cell-like cells and further expands the number of up-regulated hair cell genes. We show that the chromatin of hair cell gene loci in utricle supporting cells is maintained in a more accessible state than their counterparts in the mature cochlea, and that Atoh1 transduction of supporting cells can render the chromatin of some hair cell gene loci more accessible. However, Atoh1 transduction is unable to achieve complete conversion of supporting cells to hair cells, and we find that genes associated with mature hair cells are under-represented in our reprogrammed supporting cells. This suggests that in addition to Atoh1, other transcriptional effectors are necessary to fully reprogram supporting cells into hair cells.

## Results

### Identification of hair cell- and supporting cell-specific transcripts in the adult utricle by RNA-seq

As a first step to understanding the transcriptional responses of mature utricle supporting cells during injury and regeneration, we assembled transcriptional profiles of hair cells and supporting cells from the intact utricle. We crossed *Atoh1-CreERT2* mice (*Machold and Fishell, 2005*) with Ai3 Cre reporter mice (*Madisen et al., 2010*) and delivered tamoxifen from P10 to P14 to label hair cells with EYFP (*Figure 1A*). Three weeks later, we dissected the labeled utricles and used antibodies to GFP and Myosin7a to show that approximately 80% of utricle hair cells were labeled by this approach (*Figure 1—figure supplement 1A*). This allowed us to sort EYFP$^+$ hair cells for RNA-seq analysis (*Figure 1A*). Flow cytometric analysis of the purified hair cell population with markers of supporting cells showed they contained fewer than 1% supporting cells (*Figure 1—figure supplement 2A*; *Figure 1—figure supplement 2B*). To isolate utricle supporting cells, we made use of the fact that CD326, a 40 kDa mouse EpCAM glycoprotein is expressed by both utricle hair cells and supporting cells but not underlying stromal cells (*Hertzano et al., 2011*; *Sinkkonen et al., 2011*) (*Figure 1—figure supplement 1B*). To separate supporting cells from hair cells and stromal cells, we crossed *Gfi1-Cre* mice (*Yang et al., 2010*) with Ai3 Cre reporter mice to label hair cells with EYFP, then labeled dissociated cells from *Gfi1-Cre; Ai3* utricles with CD326 antibodies and sorted CD326 +, EYFP- supporting cells for RNA-seq analysis (*Figure 1B*). Flow cytometric analysis of the purified supporting cell population showed a complete absence of EYFP +hair cells (*Figure 1—figure supplement 2C*; *Figure 1—figure supplement 2D*).

Following RNA-seq analysis of the purified utricle hair cells and supporting cells, we performed differential expression analysis and identified 1912 transcripts more than four-fold enriched in utricle hair cells and 2619 transcripts more than four-fold enriched in supporting cells (adj. p<0.05; *Figure 1C*). To validate our gene lists, we compared our lists of enriched hair cell and supporting cells genes with genes identified in previous studies (*Figure 1—source data 1*). 70 of the top 100 enriched utricle hair cell genes have been previously reported in transcriptomic or proteomic studies of hair cells (*Cai et al., 2015*; *Scheffer et al., 2015*; *Hickox et al., 2017*). Although the transcriptome of supporting cells has been less well-characterized than that of hair cells, of our top 100 supporting cell-enriched genes, 47 have previously been shown to be expressed in cochlear supporting cells (*Maass et al., 2016* ; *Figure 1—source data 1*).

### Ad5 adenovirus specifically infects supporting cells in the adult vestibular system

Overexpression of *Atoh1* in the damaged utricle leads to the formation of new hair cell-like cells (*Shou et al., 2003*; *Taylor et al., 2018*). To understand how supporting cells respond to hair cell loss and transduction of *Atoh1*, we developed an adenoviral system to specifically infect and label supporting cells. We made adenoviral serotype 5 (dE1) vectors carrying *tdTomato*, a red fluorescent protein, under the control of the EF1α promoter (*Figure 2A*). We chose this serotype as previous studies showed it specifically infects adult utricle supporting cells (*Brandon et al., 2012*). A second vector also incorporated a full length *Atoh1* cDNA, with a self-cleaving picornavirus T2A peptide sequence inserted between *tdTomato* and *Atoh1* to make two functional proteins. 293T cells were transduced with control virus (Ad-tdTomato) and Atoh1 virus (Ad-tdTomato-Atoh1) to verify their expression. Two days after transduction, we performed immunostaining and Western blot analysis to verify that 293T cells successfully expressed the desired proteins (*Figure 2B and C*). To confirm our adenoviruses specifically targeted supporting cells, we infected mature utricle explants from 1 to 2 month old mice with Ad-tdTomato. Two days after infection, utricle explants were fixed and stained with antibodies for Myo7a and Sox2 (*Figure 2D*). Almost none of the infected cells expressed the hair cell marker Myo7a; out of two explants analyzed, we observed only a single tdTomato expressing cell co-expressing Myo7a protein, indicating our adenovirus specifically infects supporting cells in mature utricle explants.

The mature mammalian utricle undergoes very little spontaneous hair cell regeneration (*Forge, 1985*; *Forge et al., 1993*; *Li et al., 1995*; *Rubel et al., 1995*; *Lambert et al., 1997*; *Li and Forge, 1997*; *Forge et al., 1998*; *Kirkegaard and Jørgensen, 2000*), and expression of Atoh1 in

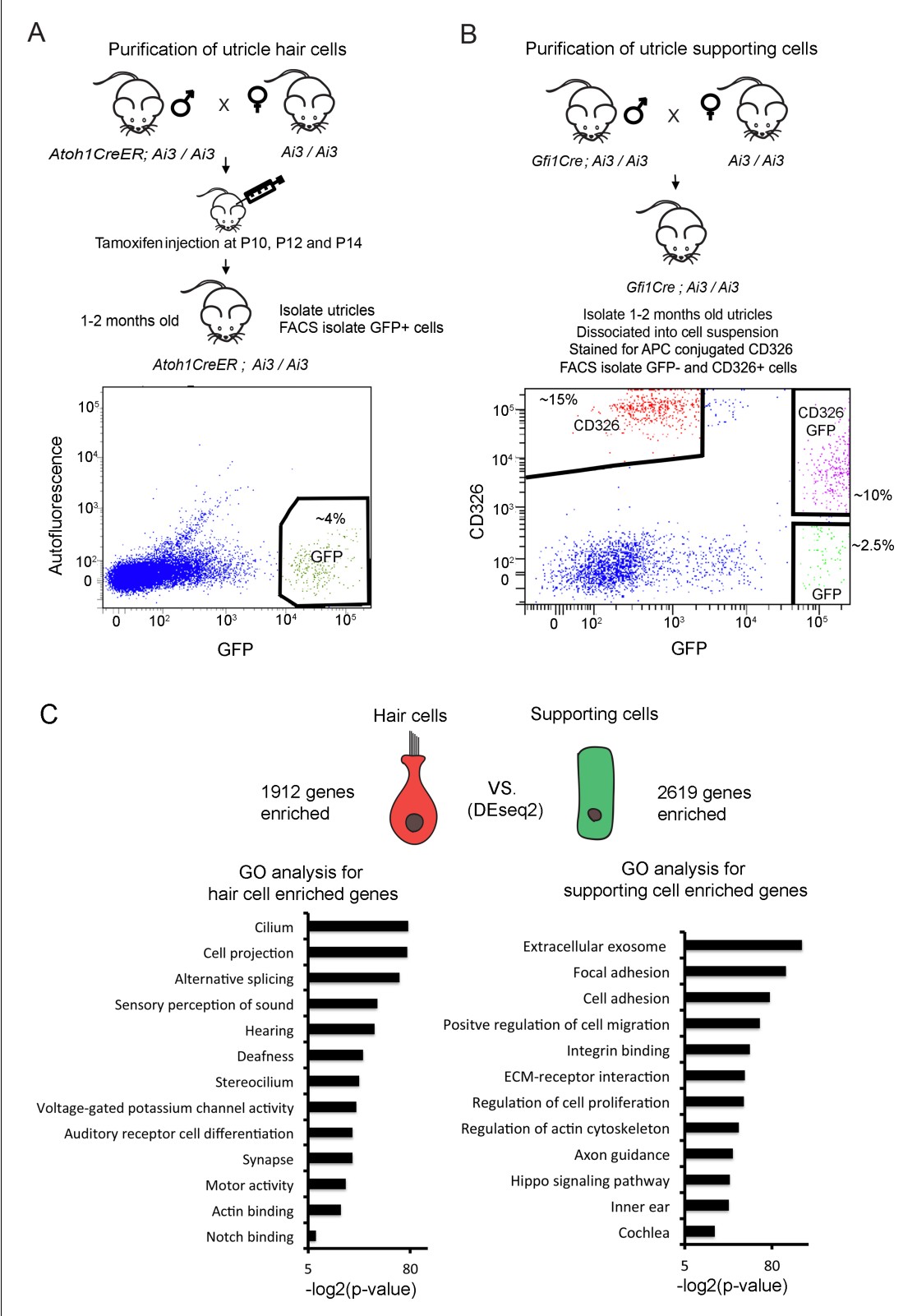

**Figure 1.** Identification of unique utricle hair cell and supporting cell transcripts by FACS sorting and RNA-sequencing. Diagrams of the breeding and FACS purification strategy to isolate utricle hair cells and supporting cells. (**A**) To isolate utricle hair cells, Ai3 reporter mice carrying an *Atoh1-CreER* transgene received tamoxifen injections at 10, 12 and 14 days after birth. GFP-expressing hair cells were sorted from 1 to 2 month old animals. (**B**) To isolate supporting cells, Ai3 reporter mice carrying a *Gfi1-Cre* transgene were sacrificed at 1–2 months of age, and the dissociated cells labeled with
*Figure 1 continued on next page*

*Figure 1 continued*

anti-CD326 conjugated to allophycocyanin (APC). Supporting cells were isolated by sorting the CD326+, GFP- population (red). (C) Identification of differentially expressed genes in utricle hair cells and supporting cells using DEseq2. Differentially expressed genes were determined based on fold change >4 and adjusted P value < 0.05. Gene ontology analysis of (GO) analysis of enriched pathways or keywords in each cell population was performed with the DAVID analysis tool using 1912 hair cell-enriched genes and 2619 supporting cell-enriched genes.
DOI: https://doi.org/10.7554/eLife.44328.002

The following source data and figure supplements are available for figure 1:

**Source data 1.** Lists of top 100 utricle hair cell-enriched genes and top 100 utricle supporting cell-enriched genes.
DOI: https://doi.org/10.7554/eLife.44328.005
**Figure supplement 1.** Fluorescence labeling of endogenous epithelial cells for RNA-seq and ATAC-seq.
DOI: https://doi.org/10.7554/eLife.44328.003
**Figure supplement 2.** Flow cytometric analysis of sorted utricle hair cells and supporting cells.
DOI: https://doi.org/10.7554/eLife.44328.004

undamaged adult mouse utricles does not generate significant numbers of hair cells (*Gao et al., 2016*). Therefore, to study the mechanism of spontaneous regeneration after hair cell death, we established a utricle organ culture system to kill hair cells. We treated utricle explants with different concentrations of gentamicin (0.1 mM, 0.5 mM, 1 mM and 2 mM) for 24 hr. We washed the explants with fresh medium, and cultured the explants for a further five days without gentamicin. 24 hr of exposure to gentamicin led to significant hair cell loss in the utricle in a dose-dependent manner (*Figure 2—figure supplement 1A*). We observed that supporting cells from cultures treated with 1 mM gentamicin for 24 hr had an abnormal, elongated morphology after 5 days (*Figure 2—figure supplement 1B*). In contrast, cultures treated with 0.1 mM gentamicin for 24 hr had lost significant numbers of hair cells after 5 days, but the remaining supporting cells retained a normal morphology (*Figure 2—figure supplement 1B*). Treatment with 0.1 mM gentamicin initiated apoptosis within 24 hr, measured by incubation with Caspase-3 enzyme substrate (*Figure 2—figure supplement 2A–C*; *Cunningham et al., 2002*). We therefore used 0.1 mM gentamicin applied for 24 hr in all subsequent experiments.

To confirm that our Atoh1 virus could induce utricle supporting cells to trans-differentiate into hair cells after hair cell death, we infected supporting cells with Ad-tdTomato or Ad-Atoh1-tdTomato one day after gentamicin treatment. In cultures infected with Ad-Atoh1-tdTomato virus, we began to observe clear Myo7a staining in about 45% of infected supporting cells five days after transduction (*Figure 2E*). Although the number of infected cells varied between each cultured utricle after ten days, we typically saw between 100 and 600 tdTomato +cells in our cultures. Ten days after transduction with Ad-Atoh1-tdTomato virus, more than 80% of the infected cells expressed Myo7a (*Figure 2E–G*), while cells infected with Ad-tdTomato virus remained Myo7a negative ten days after infection (*Figure 2—figure supplement 1B*). Our virally infected cells could be purified by FACS sorting for the tdTomato reporter (*Figure 2H*; *Figure 2I*). RT-qPCR validation showed that tdTomato transcripts were enriched 20 fold in the tdTomato positive supporting cells when compared with tdTomato negative supporting cells (*Figure 2J*) and Atoh1 transcripts were only enriched in the tdTomato positive supporting cells isolated from Ad-tdTomato-Atoh1 infected explants (*Figure 2J*).

## Utricle supporting cells up-regulate hair cell genes following hair cell death

Mammalian utricle supporting cells are able to undergo a limited degree of spontaneous regeneration to form hair cell-like cells after hair cell death (*Forge et al., 1993*; *Rubel et al., 1995*; *Kuntz and Oesterle, 1998*; *Ogata et al., 1999*; *Kawamoto et al., 2009*; *Lin et al., 2011*; *Golub et al., 2012*; *Bucks et al., 2017*). Moreover, ectopic expression of *Atoh1* can induce formation of hair cell-like cells in the mature utricle and the neonatal cochlea (*Zheng and Gao, 2000*; *Kawamoto et al., 2003*; *Shou et al., 2003*; *Izumikawa et al., 2005*; *Zhao et al., 2011*; *Kelly et al., 2012*; *Yang et al., 2012a*; *Yang et al., 2012b*; *Chen et al., 2013*; *Atkinson et al., 2014*; *Kuo et al., 2015*; *Gao et al., 2016*; *Lee et al., 2017*; *Taylor et al., 2018*). To date, however, the transcriptional response of utricle supporting cells to damage versus *Atoh1* transduction has not been evaluated.

To test the extent to which supporting cells up-regulate hair cell genes in our gentamicin-damaged utricle organ cultures, and whether this can be further enhanced by *Atoh1*, we infected 4 week

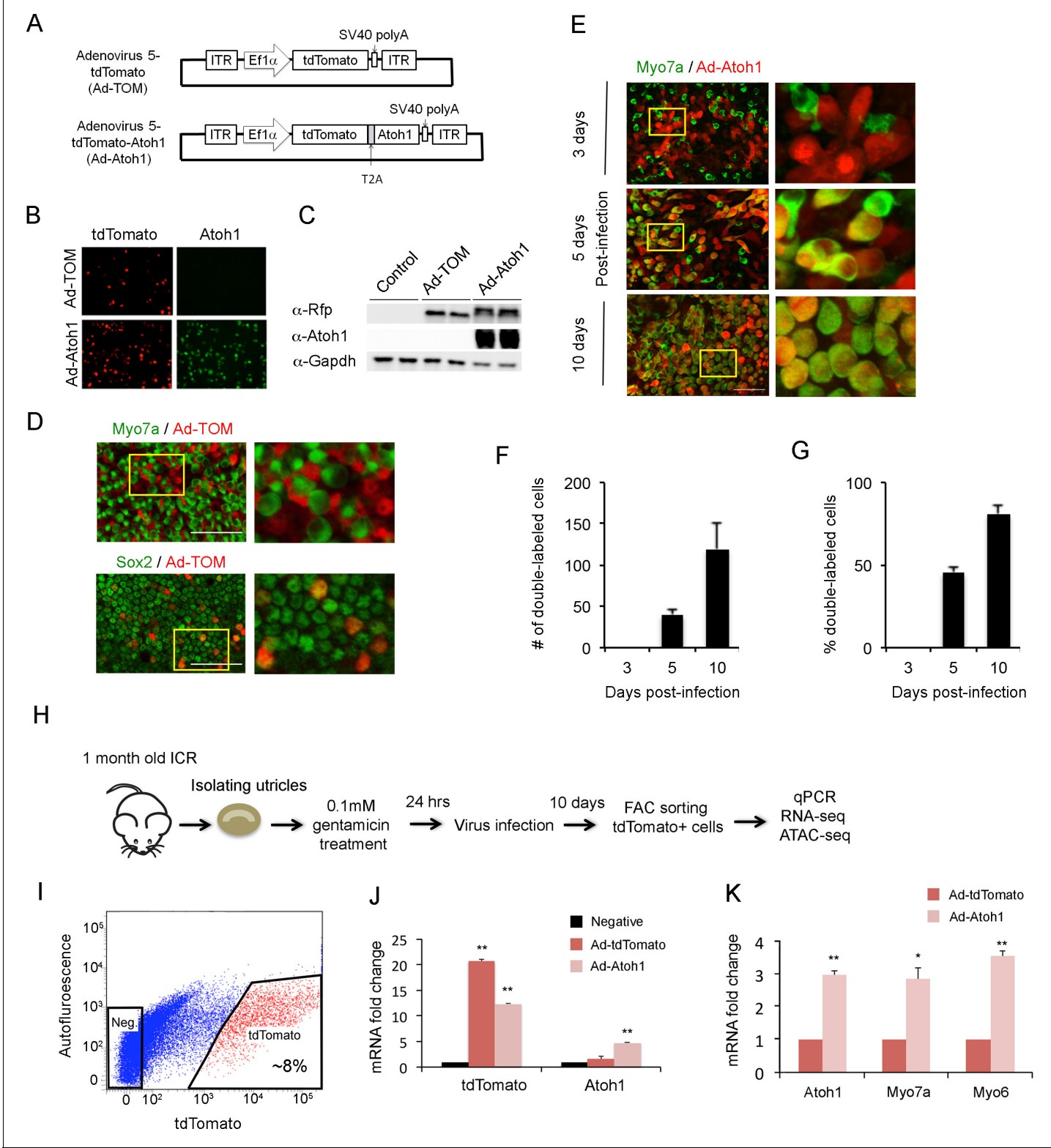

**Figure 2.** Overexpression of Atoh1 induces hair cell-like cells in the adult utricle. (**A**) Diagram of adenoviral vector constructs expressing tdTomato (Ad-TOM) and tdTomato and Atoh1 (Ad-Atoh1). ITR: inverted terminal repeat sequences, T2A: picornavirus T2A sequence. (**B**) Immunofluorescence staining for tdTomato and Atoh1 protein in 293 T cells infected with Ad-TOM or Ad-Atoh1 viruses. Scale bar, 50 μm. (**C**) Western blots for expression of tdTomato in 293 T cells infected with Ad-TOM or Ad-Atoh1 viruses. Duplicate lanes are shown for each condition. (**D**) Immunofluorescence staining for hair cell (Myo7a-cytoplasmic staining) and supporting cell (Sox2-nuclear staining) markers on utricle explants treated with Ad-tdTomato. After 48 hr, no

*Figure 2 continued on next page*

*Figure 2 continued*

tdTomato +cells express Myo7a, whereas all tdTomato +cells express Sox2. Optical sections are taken through the cell body at the level of the nucleus. Scale bar, 100 µm. (E) Infection of supporting cells with Ad-Atoh1 virus induces expression of the hair cell marker Myosin7a. Utricle cultures were examined 3, 5 and 10 days post-infection, with TdTomato marking the infected cells. The insets show enlarged images of the enclosed region by the yellow square. Scale bar, 50 µm. (F) Number of tdTomato +cells per 40x field (42,750 um$^2$) expressing Myo7a increases over the 10 day culture period. Error bars show mean ± SEM (n ≥ 6 utricles). (G) Percentage of tdTomato +cells expressing Myo7a increase over the 10 day culture period. Error bars show mean ± SEM (n ≥ 6 utricles). (H) Schematic diagram of experimental procedure for obtaining tdTomato- and Atoh1-expressing cells for RNA-seq and ATAC-seq. (I) Sample FACS profiles showing the distribution of tdTomato positive cells (red) in the virally-infected utricle cultures. (J) Q-PCR analysis of infected cells shows robust expression of tdTomato with both Ad-TOM and Ad-Atoh1 viruses, but expression of Atoh1 only in Ad-Atoh1-infected cultures. Error bars show mean ± SEM (n = 3) (*p<0.05; **p<0.01 compared with negative control) (K) Hair cell transcripts (Myo7a and Myo6) are up-regulated in Ad-Atoh1-infected cultures, but not control (Ad-TOM) cultures (n = 3) (*p<0.05; **p<0.01).

DOI: https://doi.org/10.7554/eLife.44328.006

The following figure supplements are available for figure 2:

**Figure supplement 1.** Ad-tdTomato specifically labels supporting cells after gentamicin killing.

DOI: https://doi.org/10.7554/eLife.44328.007

**Figure supplement 2.** Characterization of cell death in utricle explants acutely treated with gentamicin.

DOI: https://doi.org/10.7554/eLife.44328.008

old utricles with either Ad-tdTomato or Ad-tdTomato-Atoh1 virus, purified the infected TdTomato + cells by FACS and compared the transcriptomes of each group to the transcriptomes of endogenous utricle hair cells and supporting cells (*Figure 1C*). As expected, principal component analysis showed that Ad-tdTomato infected cells occupied an intermediate transcriptional space between utricle hair cells and supporting cells, and that Ad-tdTomato-Atoh1 infected cells more closely resembled endogenous utricle hair cells (*Figure 3A*).

We found that Ad-tdTomato-infected supporting cells up-regulated significant numbers of hair cell genes after a brief treatment with gentamicin followed by culture for 10 days (*Figure 3B*). We identified 599 hair cell transcripts that were up-regulated after this treatment. We used gene ontology (GO) analysis to describe the functional characteristics of genes up-regulated by culture and hair cell loss. Although most of the up-regulated hair cell genes have not been extensively characterized to date, top hits included gene sets for DNA repair (GO:0006281; 24 genes, p=5.02E-6), cytoskeleton (GO:0005856; 53 genes, p=1.11E-5) and cell projection (GO:0042995; 36 genes, p=1.21E-4; *Figure 3C*).

## Overexpression of *Atoh1* enhances supporting cell trans-differentiation into hair cell-like cells

Our results suggest that utricle supporting cells first induce genes involved in hair cell repair and remodeling after hair cell damage, but stop short of overt trans-differentiation to hair cell-like cells, as revealed by their failure to express markers such as Myosin7a. As expected, most of the genes up-regulated by damage were also up-regulated in cultures that received Ad-tdTomato-Atoh1 virus (*Figure 3B*, 454 genes). However, transduction of utricle cultures with *Atoh1* induced an additional 346 hair cell genes (*Figure 3B,C*). 112 of these genes appear to be generic markers of hair cells, as they are also expressed in neonatal cochlear hair cells (*Cai et al., 2015*; *Figure 3—figure supplement 1A*). The up-regulation of 35 of these hair cell genes by Atoh1 is shown in *Figure 3—figure supplement 1B*. GO analysis of the 346 additional hair cell genes induced in utricle supporting cells by Atoh1 revealed several significantly enriched pathways including gene sets for cilium (GO:0005929; 12 genes, p=8.4E-4), cilium movement (GO:0003341; five genes, p=0.0012) and syntaxin binding (GO:0019905, six genes, p=0.004). Taken together, our results suggest that Atoh1 is sufficient to activate additional hair cell genes in adult utricle supporting cells compared to those induced by hair cell death and culture alone.

Although many hair cell genes were up-regulated in supporting cells after damage and *Atoh1* transduction, many other utricle hair cell genes (967 out of 1912 genes) were not significantly up-regulated in utricle supporting cells in either of these conditions (*Figure 3B,C*). We performed GO analysis for these hair cell genes and found significant numbers of genes associated with maturation or function of hair cells. Many of these genes were associated with the GO terms 'stereocilium' (GO:0032420; 11 genes, p=4.74E-7), 'detection of mechanical stimulus involved in sensory

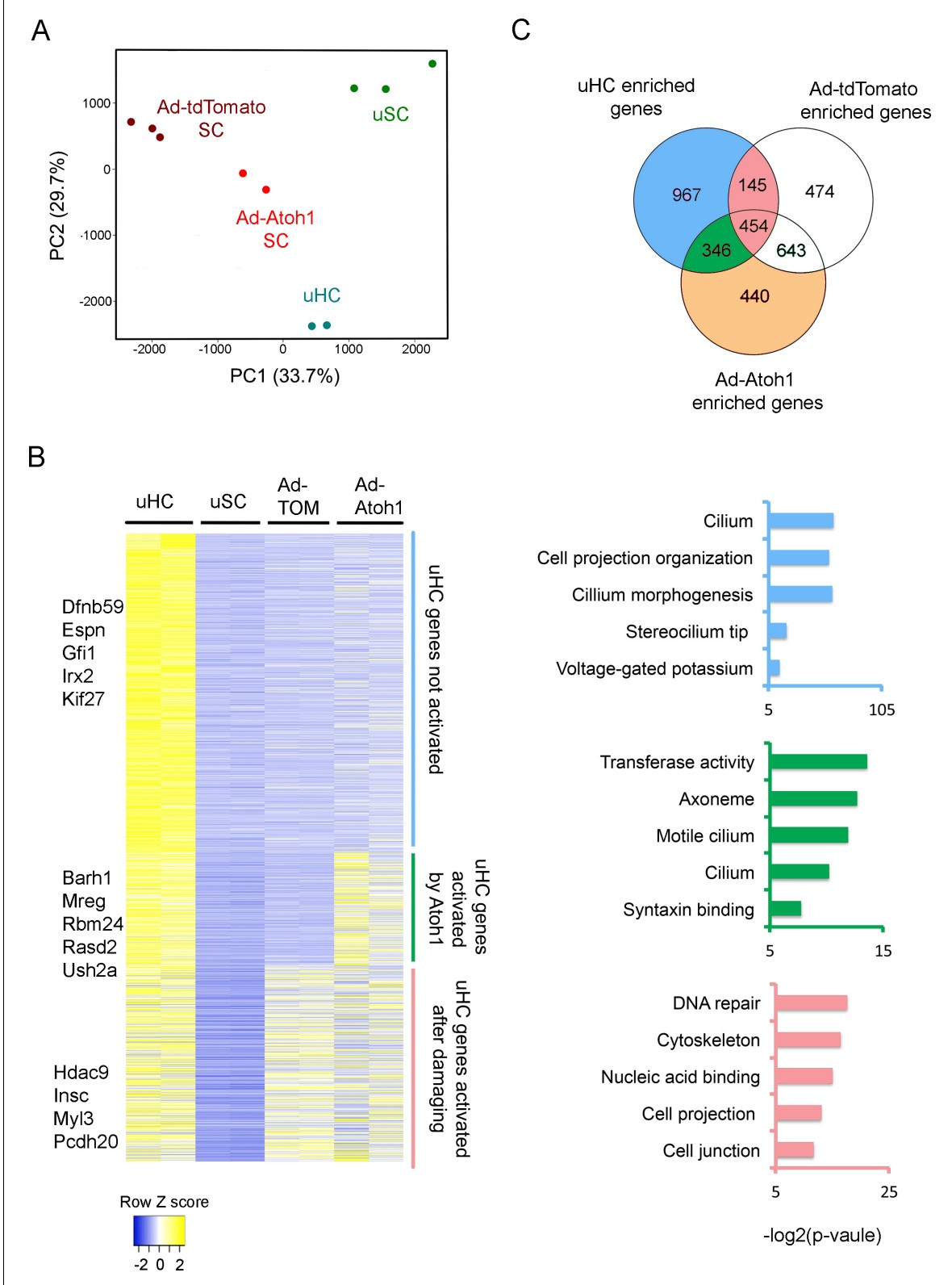

**Figure 3.** Activation of hair cell enriched genes and cell cycle genes in supporting cells after damaging to vestibular epithelium. (A) Principle component analysis (PCA) plot using the normalized counts for the RNA-seq data from endogenous utricle hair cells, utricle supporting cell, or utricle supporting cells infected with Ad-tdTomato or Ad-Atoh1. The PCA explain 33.7% (PC1) and 29.7% (PC2) of total variance. (B) Heatmap comparing gene expression level between different groups for all hair cell enriched genes. Genes were divided into three group based on their relative expression in

*Figure 3 continued on next page*

*Figure 3 continued*

viral infected groups compared with supporting cells. Gene sets were divided into three groups: Hair cell genes no activated in supporting cells; hair cells genes activated by damaging and transduction with Atoh1; hair cell genes activated by damage alone. Examples of known genes are indicated for each group. (C) Venn diagram comparing genes enriched in hair cells and genes enriched in either Ad-tdTomato or Ad-Atoh1 infected supporting cells. Gene ontology analysis of enriched pathways or keywords using genes enriched only in Ad-Atoh1 infected cells (green), enriched in Ad-tdTomato infected cells (pink) and only enriched in utricle hair cells (purple).

DOI: https://doi.org/10.7554/eLife.44328.009

The following figure supplements are available for figure 3:

**Figure supplement 1.** Validation of genes up-regulated in Ad-Atoh1 treated supporting cells.

DOI: https://doi.org/10.7554/eLife.44328.010

**Figure supplement 2.** Down-regulation of supporting cell genes in Atoh1 infected supporting cells.

DOI: https://doi.org/10.7554/eLife.44328.011

---

perception of sound' (GO:0050910; six genes, p=4.46E-4) and 'auditory receptor cell stereocilium organization' (GO:0060088; seven genes, p=6.86E-6). We also compared our gene list to the proteins that have been previously identified in the stereociliary bundle of chick vestibular hair cells (*Shin et al., 2013*). 38 of the genes coding for chick utricle hair bundle-enriched proteins were also present in our purified utricle hair cells. Some of these hair bundle genes were also enriched in either Ad-tdTomato (11 genes) or Ad-tdTomato-Atoh1 (10 genes) treated cells. However, most previously characterized hair bundle genes failed to be induced in either condition, suggesting that although Atoh1 can up-regulate many hair cell genes, it does not appear to promote expression of genes associated with the mature characteristics of hair cells. Accordingly, although our Atoh1-infected cells expressed markers such as Myosin7a, we were unable to observe obvious organization of apical actin filaments into stereocilia-like processes by fluorescence microscopy after 10 days of culture.

We also observed that infection of supporting cells with Ad-tdTomato-Atoh1 virus up-regulated a significant number of genes (440 genes) that were neither present in mature hair cells, nor up-regulated by hair cell damage and culture alone (*Figure 3C*). Since Atoh1 also functions as a transcription factor in other cell types such as cerebellar granule cells, intestinal secretory cells and Merkel cells of the skin, we asked if any of the 440 genes activated by Atoh1 were also enriched in other Atoh1-expressing cells by examining published data sets from these tissues. Although some of these 440 genes were indeed expressed in cerebellar granule cells (*Klisch et al., 2011*), gut secretory cells (*Lo et al., 2017*) or Merkel cells (*Haeberle et al., 2004*), we also found some of the 440 genes to be expressed in cell populations that do not express Atoh1, such as smooth muscle (*Himes et al., 2014*) and kidney (*Habuka et al., 2014*) in similar proportions to the Atoh1-expressing tissues (*Figure 3—figure supplement 1C*). Thus, it is unlikely that Atoh1 transduction is specifically up-regulating Atoh1 gene regulatory networks that normally function in other Atoh1-expressing cell types. Rather, it is more likely that Atoh1 is activating cryptic expression of genes that happen to contain accessible target sites in utricle supporting cells.

The transformation of supporting cells into hair cells not only involves the up-regulation of hair cell genes, but must also require the down-regulation of supporting cell genes. To determine the extent to which supporting cell genes are down-regulated after hair cell killing and culture, and whether this down-regulation can be enhanced by Atoh1, we analyzed the behavior of supporting cell-enriched genes in our RNA-seq experiments. We generated heat maps to observe the behavior of 2619 supporting cell-enriched genes in utricle supporting cells following hair cell killing and culture after infection with either Ad-tdTomato or Ad-tdTomato-Atoh1 virus. Several hundred supporting cell genes (608 were significantly down regulated in utricle supporting cells after hair cell killing and culture (*Figure 3—figure supplement 2A*), including some genes associated with differentiation (*Figure 3—figure supplement 2B*). A small number (107) of additional supporting cell genes were down-regulated when utricle supporting cells were infected with Atoh1 virus (*Figure 3—figure supplement 2A*). Significantly, many supporting cell genes remained expressed in our cultured utricles after 10 days. Gene ontology analysis indicated that a significant number of genes remaining expressed in supporting cells were associated with GO terms such as cell adhesion, focal adhesion and integrin binding (*Figure 3—figure supplement 2B*).

## Single cell RNA-seq analysis of the response of supporting cells to hair cell death and Atoh1 transduction

Our bulk RNA-seq analysis reveals that utricle supporting cells collectively up-regulate many hair cell genes after the utricle is treated with gentamicin and cultured for 10 days, and that additional hair cell genes are up-regulated by transduction with *Atoh1*. To determine to what extent our bulk RNA-seq data represents the averaging of varying responses of supporting cells to these conditions, we performed single cell RNA-seq analysis of purified supporting cells from utricles treated with genta-micin, infected with Ad-tdTomato or Ad-tdTomato-Atoh1 and cultured for 10 days (the same proto-col as described in *Figure 2H*). To understand the transcriptional responses of individual cells in depth, we used the MATQ-seq technique (*Sheng et al., 2017*) that offers the advantages of a greater read depth and representation of transcripts than conventional single cell RNA-seq proto-cols. We manually picked, amplified and sequenced RNA from 34 individual tdTomato and 27 tdTo-mato-Atoh1 transduced supporting cells. We analyzed cells that had more than 1 million uniquely mapped reads, with each cell having around fifteen thousand genes identified. Of those 61 cells, 34 individual tdTomato and 22 tdTomato-Atoh1 transduced supporting cells passed quality control tests and were analyzed further.

To group single cells into respective cell types, we performed graph-based clustering followed by visualization using t-distributed stochastic neighbor embedding. Next, all 56 individual transcrip-tomes were computationally analyzed and the results were visualized after T-distributed stochastic neighbor embedding (t-SNE) dimensionality reduction (*Figure 4A*). We found that the two groups of infected cells were generally well separated from each other in transcriptional space (*Figure 4A*). We performed an unsupervised hierarchical clustering analysis of the 56 cells (*Figure 4C*) on the basis of their expression of the top 500 hair cell-specific genes identified in our initial bulk RNA-seq analysis of mature utricle hair cells (*Figure 4C*). Although we observed a heterogeneity in the relative expression of hair cell genes across individual cells, tdTomato-Atoh1 and tdTomato transduced sup-porting cells tended to cluster separately. Of four sub-clusters identified, one contained exclusively Atoh1 transduced cells (n = 11 cells; *Figure 4C*, red cluster) and two clusters contained either exclu-sively tdTomato transduced cells (n = 8 cells; *Figure 4C*; dark blue cluster) or almost all tdTomato transduced cells (n = 20 tdTomato cells and n = 1 Atoh1 cell; *Figure 4C*, light blue cluster). A final cluster was more heterogeneous, containing 10 Atoh1 cells and six tdTomato cells (*Figure 4C*, orange cluster). Many previously identified hair cell genes were expressed at higher levels in Atoh1-transduced cells compared with TdTomato alone – for example *Pou4f3, Gfi1, Tmc1, Lhx3, Pvalb, Tomt, Mgat5b, Mreg* and *Rab15* (*Cai et al., 2015*; *Figure 4B*, *Figure 4—figure supplement 1*). However, we also saw a subset of hair cell genes that did not show significant differences between Atoh1- and tdTomato-transduced cells (for example, *Fscn2*; *Figure 4B,C*; *Figure 4—figure supple-ment 1*).

The up-regulation of hair cell genes as supporting cells trans-differentiate into hair cell-like cells is likely to be accompanied by a concomitant down-regulation of supporting cell genes. To test this, we performed a second hierarchical clustering of the single Atoh1 and TdTomato-transduced sup-porting cells using the expression of the top 500 supporting cell-specific genes identified in purified utricle supporting cells (*Figure 4—figure supplement 2*). As expected, supporting cells that were transduced with Atoh1 tended to express lower levels of the top 500 supporting cell genes com-pared to supporting cells that received tdTomato virus alone.

## Differences in the regenerative ability of utricle and cochlear supporting cells correlate with chromatin accessibility at the enhancers of hair cell genes

The adult utricle is able to undergo a limited amount of regeneration in response to hair cell loss, whereas this has not been observed in the adult cochlea (*Forge et al., 1993*; *Rubel et al., 1995*; *Kuntz and Oesterle, 1998*; *Ogata et al., 1999*; *Kawamoto et al., 2009*; *Lin et al., 2011*; *Golub et al., 2012*). Moreover, our data, together with previous studies, show that Atoh1 transduc-tion is able to promote some degree of supporting cell trans-differentiation into hair cell-like cells in the adult utricle, but not in the adult cochlea (*Shou et al., 2003*; *Kelly et al., 2012*; *Liu et al., 2012*; *Gao et al., 2016*; *Taylor et al., 2018*). One explanation for these differences in regenerative capac-ity is that loci of hair cell genes may be more transcriptionally accessible in utricle supporting cells

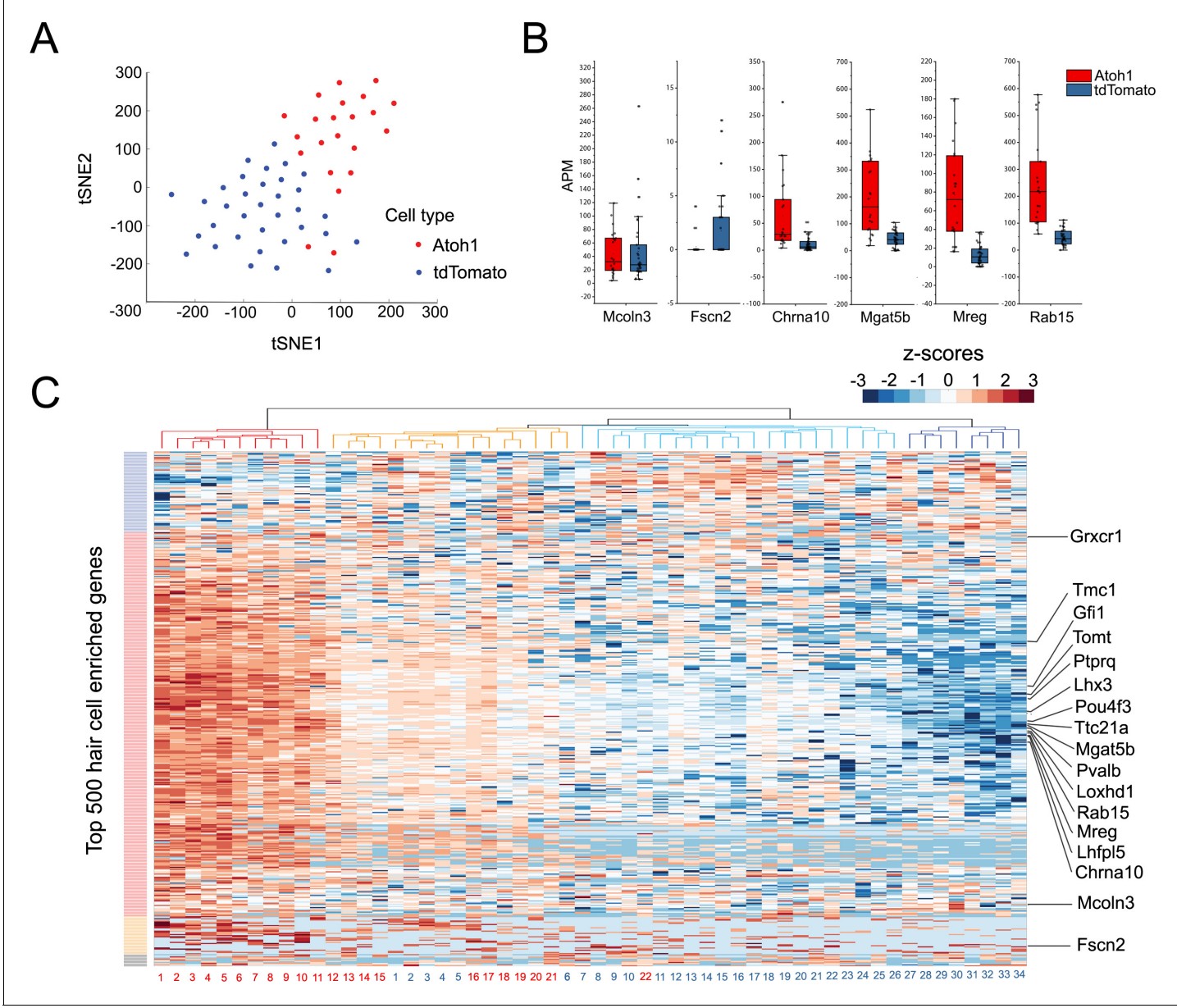

**Figure 4.** Single Cell RNA-seq analysis reveals heterogeneity in the response of supporting cells to hair cell death and Atoh1 transduction. (**A**) t-SNE analysis of single cells (*n* = 56) for Ad-tdTomato (n = 34) and Ad-tdTomato-Atoh1(n = 22) infected supporting cells using MATLAB. (**B**) Box-plot for verified hair cell specific genes, including hair cell genes that are not induced by Atoh1 virus infection (Mcoln3, Fscn2) and hair cell genes that are induced by Atoh1 virus infection (Chrna10, Mgat5b, Mreg, Rab15). APM: amplicon per million amplicons. (**C**) Heatmap showed hierarchical clustering of gene expression level between each single cells (Ad-tdTomato-Atoh1 infected cells labeled in red, and Ad-tdTomato infected supporting cell labeled in blue) for top 500 utricle hair cell enriched genes. Color code on the left suggested clustered genes population while on the top showing how each cells clustered together. Many hair cell genes that are previously identified are also showed in the heatmap.

DOI: https://doi.org/10.7554/eLife.44328.012

The following figure supplements are available for figure 4:

**Figure supplement 1.** Ad-Atoh1 infected cells showed increased transcripts for many but not all verified hair cell genes.
DOI: https://doi.org/10.7554/eLife.44328.013
**Figure supplement 2.** Single Cell RNA-seq analysis reveals Atoh1 transduction down-regulate supporting cell enriched genes.
DOI: https://doi.org/10.7554/eLife.44328.014

compared to their cochlear counterparts. To test this, we used ATAC-seq to evaluate the differential accessibility of hair cell gene loci in different cell populations in the utricle and cochlea. For this analysis, we focused on the 1912 genes we showed to be enriched in utricle hair cells (*Figure 1C*). As expected, ATAC-seq identified many examples of both proximal elements corresponding to the transcriptional start site (TSS) and distal elements in hair cell gene loci that were accessible in hair cells but not supporting cells (*Figure 5A*), and supporting cell gene loci that were accessible in supporting cells but not hair cells (*Figure 5B*). However, we also observed the presence of accessible DNA, indicative of relatively open chromatin, around the transcriptional start sites, intergenic regions and introns of hair cell gene loci in utricle supporting cells (*Figure 5C*; *Figure 5—figure supplement 1A,B*), suggesting that adult utricle supporting cells maintain some hair cell loci in a transcriptionally accessible state. However, these hair cell loci were nevertheless expressed at 1–2 orders of magnitude lower levels in supporting cells than hair cells. Examples of such loci are shown in *Figure 5—figure supplement 2*. Moreover, we saw no correlation between the expression levels of hair cell genes in either cochlear (r = 0.04) or utricle (r = 0.02) supporting cells and their relative chromatin accessibility (*Figure 5—figure supplement 3A,B*).

We next compared our ATAC-seq data from utricle supporting cells with a similar analysis of three week old cochlear supporting cells purified from *Lfng-GFP* transgenic mice that exclusively label supporting cells in the mature cochlea (*Maass et al., 2016*). We focused our analysis on 428 genes that RNA-seq analysis showed to be expressed in both utricle and cochlear hair cells. Comparison of the ATAC-seq peaks identified around the TSS, intergenic regions and introns of these common hair cell genes – for example, the hair cell gene *Chrna10* - revealed more open peaks in adult utricle supporting cells compared to cochlear supporting cells (*Figure 5D*). Heat maps of all ATAC-seq peaks called for the common hair cell genes observed in mature utricle hair cells, utricle supporting cells and mature cochlear supporting cells are shown in *Figure 5D*. We found that many hair cell peaks in the vicinity of the transcriptional start site were similarly accessible in all three cell types. However, when we analyzed ATAC-seq peaks in distal regions more typically associated with enhancers, we saw that peaks that were accessible in mature utricle hair cells and supporting cells were comparatively less accessible in mature cochlear supporting cells (*Figure 5D*; *Figure 5—figure supplement 1C,D*). This suggests that differential chromatin accessibility may be one explanation for why mature cochlear supporting cells may not undergo spontaneous regeneration or trans-differentiation after expression of Atoh1.

Although we demonstrated that many hair cell genes can be activated in utricle supporting cells after culturing in the presence of gentamicin, and that transduction of Atoh1 can up-regulate additional hair cell genes, we nevertheless observed that almost 1000 utricle hair cell genes were not significantly up-regulated in supporting cells in either condition (*Figure 3B,C*, *Figure 6—figure supplement 1A,B*). It is possible that the chromatin at these loci was less accessible in utricle supporting cells compared to that of genes that were able to be up-regulated. To test this, we analyzed our ATAC-seq data to compare chromatin accessibility between hair cell genes that can be up-regulated in supporting cells after hair cell killing, or after hair cell killing and Atoh1 transduction (945 genes), versus hair cell genes that failed to be up-regulated in these conditions (967 genes; *Figure 6—figure supplement 1A*). We observed no significant differences in chromatin accessibility in supporting cells around the TSS, intergenic regions or introns for the 945 up-regulated hair cell genes compared to the 967 hair cell genes that were not up-regulated (*Figure 6A*; *Figure 6—figure supplement 1B*). These data suggest that relative chromatin accessibility is not likely to be a significant determinant of whether a hair cell gene can be activated in utricle supporting cells after cell death or Atoh1 transduction. Rather, it is possible that these genes are either not directly regulated by Atoh1, or that other transcription factors are required for their induction in addition to Atoh1.

Atoh1 is normally down-regulated shortly after hair cells differentiate (*Shailam et al., 1999*; *Driver et al., 2013*; *Cai et al., 2015*; *Maass et al., 2015*) and is therefore unlikely to regulate hair cell genes in mature hair cells. We analyzed ATAC-seq peaks from both utricle and cochlear hair cells and supporting cells of different ages by de novo motif analysis to determine the representation of Atoh1 binding sequences in ATAC-seq peaks from each cell type (*Figure 6—figure supplement 2*). As expected, whereas other known hair cell specific transcription factor binding motifs, including Pou4f3 and CTCF, are highly enriched in ATAC-seq peaks of adult utricle hair cells, Atoh1 binding motifs were not significantly enriched in ATAC-seq peaks of either adult utricle hair cells or supporting cells (p values = 1.00E-9 and p value = 1 respectively; *Figure 6—figure supplement 2* and

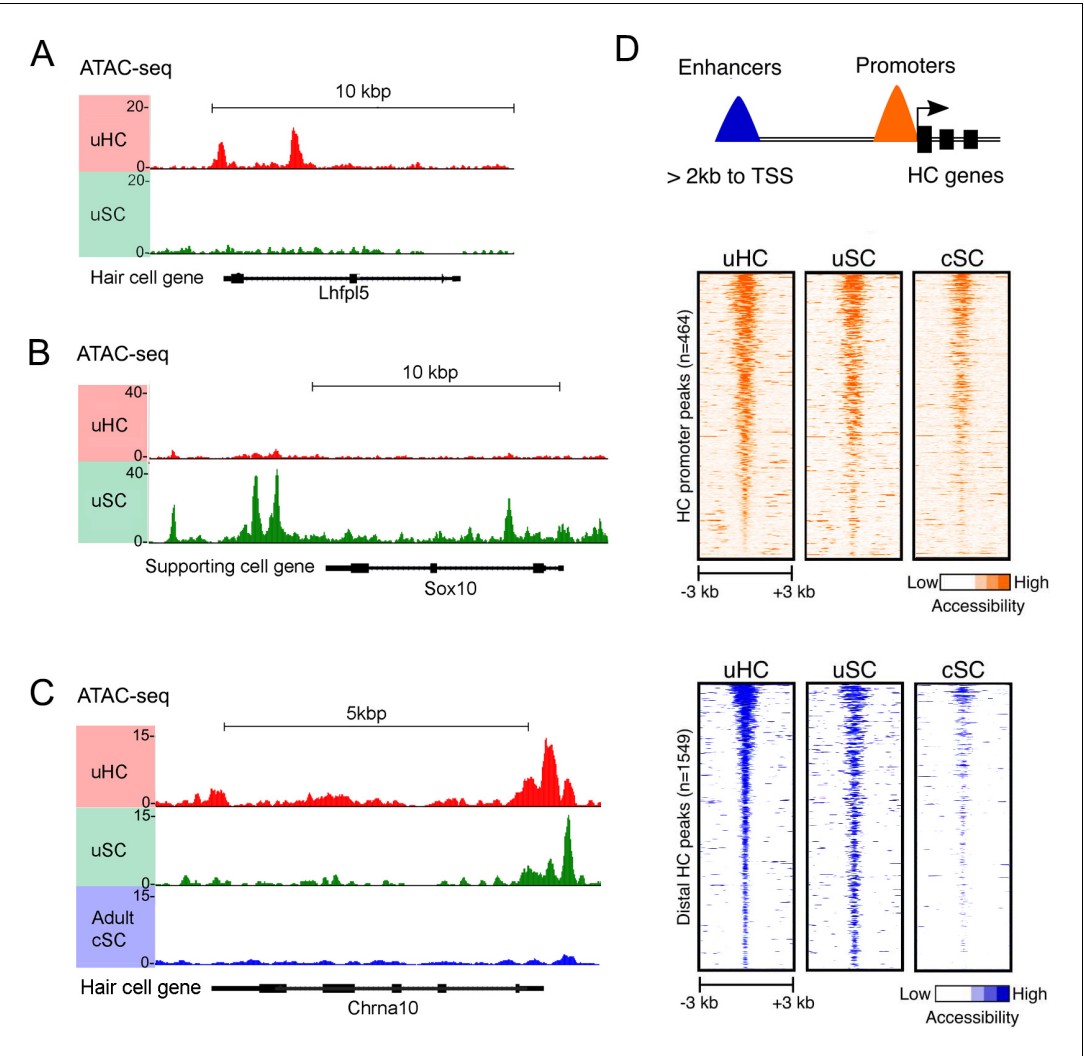

**Figure 5.** ATAC-seq analysis shows hair cell gene loci are more accessible in utricle supporting cells compared to cochlear supporting cells. (**A**) Example of ATAC-seq profiles for a hair cell-specific gene, Lhfpl5, that is not expressed in supporting cells. Peaks are present in the hair cell sample, but not the supporting cell sample **B**) Example of ATAC-seq profiles for a supporting cell-specific gene, Sox10, that is not expressed in hair cells. Peaks are present in the supporting cell sample, but not the hair cell sample. (**C**) Example of ATAC-seq profiles for a hair cell-specific gene, Chrna10, that is not expressed in supporting cells. A similar distribution of peaks is observed in utricle hair cells and supporting cells, but the peaks are absent in P21 cochlear supporting cells. (**D**) A heat map showing the correlation of ATAC-seq signals from utricle hair cells, supporting cells and P21 cochlear supporting cells. Peaks were grouped on the basis of their proximity to the transcriptional start site (TSS) or grouped as occurring in distal enhancers if they were located >2 kb away from the annotated TSS. Top, ATAC-seq signal (read depth) across putative hair cell gene promoters (n = 1549). High read intensity is shown in orange. Bottom, ATAC-seq signal (read depth) at more distal hair cell gene enhancers (n = 464). High read intensity is shown in blue.

DOI: https://doi.org/10.7554/eLife.44328.015

The following figure supplements are available for figure 5:

**Figure supplement 1.** ATAC-seq analysis shows hair cell gene loci are more accessible in utricle supporting cells compared to cochlear supporting cells.

DOI: https://doi.org/10.7554/eLife.44328.016

**Figure supplement 2.** Expression level of hair cell specific genes in utricle hair cells, supporting cells and adult cochlear supporting cells.

DOI: https://doi.org/10.7554/eLife.44328.017

**Figure supplement 3.** Global Expression and Chromatin Accessibility Correlation Analysis.

DOI: https://doi.org/10.7554/eLife.44328.018

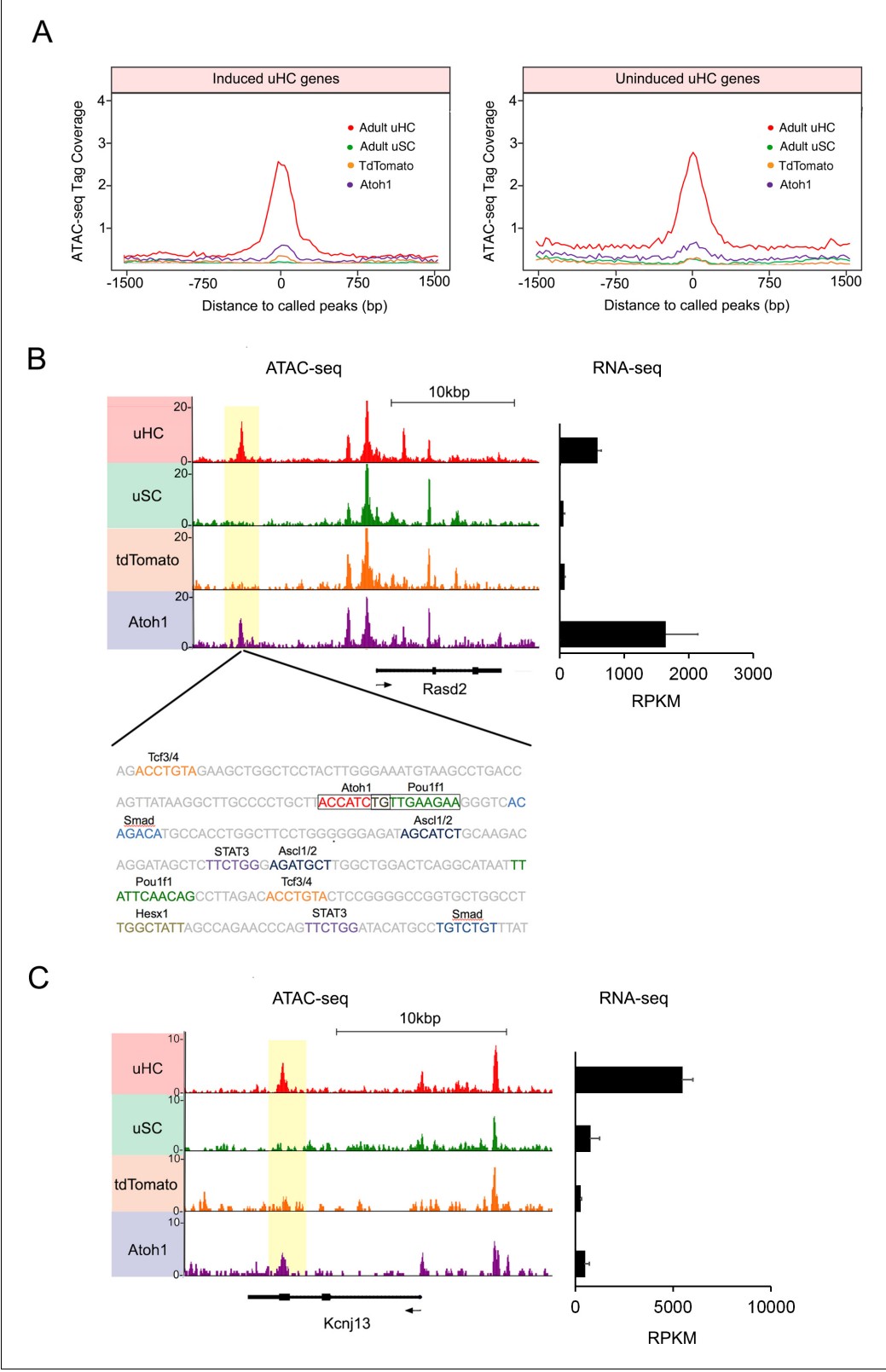

**Figure 6.** Atoh1 overexpression in utricle supporting cells can remodel chromatin of hair cell gene loci. (**A**) Aggregation plot showing enrichment of ATAC-seq signals from each condition for unique utricle peaks that are assigned to the 945 hair cell genes that can be induced in supporting cells (left) or the 967 hair cell genes that cannot be induced in supporting cells (right). No significant differences are seen in the relative accessibility of these two groups of hair cell genes. (**B**) Example of an ATAC-seq profile for a hair cell-specific gene, Rasd2 that can be altered by Atoh1 transduction. RNA-seq

*Figure 6 continued on next page*

*Figure 6 continued*

analysis shows that Rasd2 is up-regulated in supporting cells by transduction with Atoh1, but not with the TdTomato control virus. A unique ATAC-seq peak present in hair cells but not supporting cells (yellow highlighting). However, the peak re-appears in supporting cells transduced with Atoh1, but not with the TdTomato control virus. The DNA sequence corresponding to this peak shows binding sites for Atoh1 along with other transcription factors. (C) Example of ATAC-seq profiles for a hair cell-specific gene, Kcnj13. RNA-seq analysis shows that Kcnj13is not up-regulated in supporting cells by transduction with Atoh1 or with the TdTomato control virus. Nevertheless, a unique ATAC-seq peak present in hair cells is re-opened in supporting cells transduced with Atoh1.

DOI: https://doi.org/10.7554/eLife.44328.019

The following source data and figure supplements are available for figure 6:

**Source data 1.** Known motif analysis of transcription factor binding sites that are enriched in the accessible chromatin of utricle hair cells, utricle supporting cells, neonatal cochlear hair cells, Ad-tdTomato or Ad-Atoh1 infected supporting cells.
DOI: https://doi.org/10.7554/eLife.44328.024

**Figure supplement 1.** Analysis of hair cell gene up-regulation in supporting cells.
DOI: https://doi.org/10.7554/eLife.44328.020

**Figure supplement 2.** Motif enrichment analysis showing of utricle hair cells and supporting cells, and changes induced by Atoh1 transduction.
DOI: https://doi.org/10.7554/eLife.44328.021

**Figure supplement 3.** Atoh1 overexpression induced chromatin accessibility is independent for up regulation of hair cell genes in supporting cells.
DOI: https://doi.org/10.7554/eLife.44328.022

**Figure supplement 4.** Atoh1 Overexpression Promotes Ectopic Peak Activation Proximal to Hair Cell Genes.
DOI: https://doi.org/10.7554/eLife.44328.023

*Figure 6—source data 1*). In contrast, Atoh1 motifs were highly enriched in ATAC-seq peaks from neonatal cochlear hair cells where Atoh1 is still expressed (p value = 1.00E-1040; *Figure 6—figure supplement 2C*; and *Figure 6—source data 1*). These results suggest that Atoh1 binding sites are available in developing hair cells but become less accessible in the course of hair cell maturation.

## Atoh1 overexpression changes the epigenetic landscape of utricle supporting cells

In addition to regulating transcription, bHLH genes have been proposed to act as pioneer factors in certain contexts – they can bind to nucleosomes in quiescent region of chromatin and increase chromatin accessibility (*Soufi et al., 2015*). It is therefore possible that when Atoh1 is expressed in mature supporting cells, it may bind to quiescent hair cell loci and render them more transcriptionally accessible. We were able to find examples of this behavior in our ATAC-seq analysis. For example, the hair cell specific gene, *Rasd2* contains a strong ATAC-seq peak in utricle hair cells, but the peak is not present in utricle supporting cells (*Figure 6B*). However, transduction of supporting cells with Ad-tdTomato-Atoh1 but not Ad-tdTomato renders this region accessible to a comparable degree to hair cells (*Figure 6B*). Analysis of the region of DNA corresponding to this peak confirmed the presence of an Atoh1 binding site (*Figure 6B*), together with binding sites for Pou, Tcf and Ascl transcription factors, and binding sites for Smad transducers of TGFβ superfamily signaling. Motif analysis showed binding sites for other hair cell transcription factors, including Tcf12, were also enriched in peaks that were created by transduction with Ad-tdTomato-Atoh1 virus (*Figure 6—figure supplement 2C*). To see if this phenomenon extended to other loci, we performed de novo motif analysis on ATAC-seq peaks identified in utricle supporting cells that had been transduced with exogenous *Atoh1*, with tdTomato-transduced cells acting as a control. We observed that Atoh1 binding motifs were enriched in the open regions of Ad-tdTomato-Atoh1 infected cells (P value = 1.00E-107; *Figure 6—figure supplement 2C*; *Figure 6—source data 1*), but not in Ad-tdTomato infected cells (p value = 1; *Figure 6—figure supplement 2C*; *Figure 6—source data 1*).

Together, these results suggest that activation of Atoh1 in supporting cells can either directly or indirectly reprogram the epigenetic landscape of utricle supporting cells to make some hair cell genes loci more accessible. However, increased chromatin accessibility around a given gene does not necessarily imply that gene will be transcribed. An example of this is shown for the hair cell gene *Kcnj13*, where Atoh1 transduction of supporting cells restores an ATAC-seq peak found in hair cells, but does not activate transcription of the gene in the transduced supporting cells (*Figure 6C*). Moreover, we saw no changes in expression of the three genes (*Gigyf2*, *Efhd1* and *Ngef*) located within ±200 kb of *Kcnj13*. When we compared ATAC-seq peaks unique to hair cells between induced

and uninduced hair cell genes, we found about ten percent of those peaks became accessible in supporting cells after Atoh1 transduction in each group (*Figure 6—figure supplement 3A*). This shows that Atoh1 is able to cause opening of the chromatin around some hair cell gene loci, but that this opening does not necessarily lead to transcription of those genes, suggesting additional transcription factors may be required to activate transcription of a given gene. Furthermore, we observed that Atoh1 transduction also leads to the opening of several hundred peaks that are not present in utricle hair cells (*Figure 6—figure supplement 4A,B*). This raises the possibility that some of the genes induced by Atoh1 may be activated by Atoh1 engaging cryptic binding sites.

## Discussion

Vestibular supporting cells in the adult mouse undergo a very slow rate of hair cell turnover (*Bucks et al., 2017*) and do not show a significant response to Atoh1 transduction (*Gao et al., 2016*). In contrast, hair cell killing by aminoglycosides, extended cell culture or genetic ablation strategies appears to prime adult vestibular supporting cells to up-regulate some hair cell genes and generate hair cell-like cells (*Golub et al., 2012*; *Bucks et al., 2017*), and this hair cell generation can be enhanced by interventions such as Atoh1 transduction or Notch inhibition (*Shou et al., 2003*; *Golub et al., 2012*; *Gao et al., 2016*; *Taylor et al., 2018*). In the present study, we used a low dose of gentamicin applied for 24 hr that was sufficient to kill significant numbers of hair cells, but maintained supporting cells in a normal morphological state. Consistent with the idea that hair cell damage allows supporting cells to adopt a cell state that is more amenable to trans-differentiation into hair cells, we observed significant transcriptional changes in utricle supporting cells over a ten day period. However, these cells did not display any overt morphological features of hair cells, nor did they up-regulate canonical markers of hair cells such as Myosin7a. Adenoviral transduction of Atoh1 was required to induce expression of hundreds of additional hair cell genes (*Figure 3B*). It is unclear whether the 'primed' state of supporting cells persists for long periods after hair cell death, or whether it is an acute response to injury. Although our culture system is not suitable to answer such questions, it will be of interest to combine in vivo models of hair cell killing with transgenic methods of Atoh1 activation to measure for how long mature utricle supporting cells retain regenerative capacity after damage and hair cell loss.

The induction of hair cell genes in the utricle following hair cell death or Atoh1 transduction stands in contrast to the failure of the mature cochlea to generate hair cell-like cells after damage, and the failure of Atoh1 to promote trans-differentiation of mature cochlear supporting cells (*Kelly et al., 2012*; *Liu et al., 2012*). Our ATAC-seq experiments show that the chromatin around the loci of hair cell genes common to the cochlea and utricle is more accessible in utricle supporting cells compared to their cochlear counterparts, which may underlie their higher capacity for regeneration. At present, we do not know how hair cell gene loci in utricle supporting cells are maintained in a more permissive state than cochlear supporting cells. Studies in chicken and zebrafish, which exhibit spontaneous hair cell regeneration, have demonstrated that genes involved in histone remodeling are important regulators of supporting cell proliferation. In birds, HDAC inhibition in regenerating vestibular epithelium results in decreased supporting cell proliferation (*Slattery et al., 2009*). Similarly in zebrafish, HDAC inhibition result in reduced lateral line supporting cell proliferation and thus hair cell regeneration (*He et al., 2014*; *He et al., 2016*). Inhibition of lysine-specific demethylase 1 (LSD1, also known as KDM1A) in zebrafish also prevents supporting cell proliferation (*Tang et al., 2016*). Finally, a recent study showed that BMI1, part of the Polycomb repressive complex1 (PRC1), is required for supporting cell proliferation in neonatal mice by maintaining high levels of canonical Wnt signaling (*Lu et al., 2017*). Regardless of the mechanistic basis for the difference in chromatin accessibility in utricle and cochlear supporting cells, we believe it may contribute to the regenerative differences seen between these two mammalian sensory organs in mature animals.

Despite the large number of hair cell genes that could be activated in mature utricle supporting cells, we nevertheless observed almost 1000 utricle hair cell genes that were not up-regulated in supporting cells, even after hair cell killing and Atoh1 transduction. Many of these genes were associated with maturation of hair cell function, including hair bundle and kinocilium development. A recent study in which human utricle tissue was transduced with an Atoh1 adenovirus also reported phenotypes consistent with incomplete hair cell differentiation, including an absence of stereocilia and actin bundling proteins such as espin, and an absence of a cuticular plate and ribbon synapses

(*Taylor et al., 2018*). What are the reasons for this incomplete trans-differentiation of supporting cells to hair cells after transduction of Atoh1? One possibility is that our ten day period of culture and Atoh1 reprogramming is not sufficient to fully up-regulate the most mature hair cell genes. Studies of cellular reprogramming by transcription factors suggest that reprogramming events at the level of single cells initially proceed in a stochastic manner before an eventual entrainment of the fully reprogrammed state (*Hanna et al., 2009*; *Buganim et al., 2012*; *Lujan et al., 2015*). Our single cell analysis (*Figure 4*), together with a recent single cell analysis of Atoh1 activation in the cochlea (*Yamashita et al., 2018*), show a wide degree of heterogeneity in transcriptional responses to Atoh1 transduction that may underlie this incomplete transformation to hair cells. Alternatively, since Atoh1 is normally down-regulated as hair cells differentiate (*Shailam et al., 1999*; *Driver et al., 2013*; *Cai et al., 2015*; *Maass et al., 2015*), it is possible that persistent expression of Atoh1 after adenoviral infection actively represses genes associated with hair cell maturation that would normally be activated as Atoh1 levels decline. A third possibility is that some hair cell genes are regulated entirely independently of Atoh1 and are unaffected by Atoh1 expression. Alternatively, some hair cell genes may require additional transcription factors or co-activators for expression (for example, Gfi1 and Pou4f3; *Costa and Henrique, 2015*; *Costa et al., 2015*), and these are not up-regulated by Atoh1 expression alone. Finally, it is possible that the chromatin of some genes associated with hair cell maturation are less transcriptionally accessible in supporting cells than other genes associated with earlier steps in hair cell development. When we examined the behavior of supporting cell genes in response to hair cell killing and Atoh1 transduction, we also found many genes that remained expressed in utricle supporting cells despite 10 days of culture and Atoh1 expression. The reasons described above may also account for the persistent expression of these genes under these conditions. It is also possible that some of the genes expressed in supporting cells may act to directly or indirectly prevent access of Atoh1 to some of its target loci, and such genes might also constitute therapeutic targets.

The epigenetic and transcriptomic analysis undertaken in this study suggests that Atoh1 is not sufficient to induce expression of many hair cell genes in supporting cells, even when their chromatin is accessible. Our ATAC-seq analysis of Atoh1-transduced supporting cells suggests that chromatin accessibility of a given hair cell locus does not necessarily correlate with the ability of that locus to be transcriptionally activated by Atoh1 in utricle supporting cells. We compared the chromatin accessibility between hair cell genes that can be up regulated (945 genes) versus hair cell genes that failed to be activated (967 genes) in supporting cells (*Figure 6—figure supplement 1A*) but found no significant differences in the accessibility of either the transcriptional start site or enhancer regions (*Figure 6—figure supplement 1B*). Moreover, although we provide examples in which Atoh1 transduction can either directly or indirectly render certain hair cell loci more accessible when transduced into supporting cells (*Figure 6B*), we find no relationship between the opening up of chromatin at a given hair cell gene locus in response to Atoh1 and the activation of transcription from that locus in supporting cells in response to Atoh1 (*Figure 6*). Instead, de novo motif analysis of the chromatin peaks opened by Atoh1 shows that different sets of transcription factor binding sites are enriched in the peaks of loci that can be induced by Atoh1 compared to loci that cannot by induced by Atoh1 alone (*Figure 6—figure supplement 3B*).

Our data thus suggest that chromatin accessibility is not a good predictor of whether a hair cell gene is induced by Atoh1 in utricle supporting cells. Rather, we suggest that while Atoh1 may be sufficient to induce the expression of some hair cell genes, additional transcription factors or chromatin remodelers may be necessary to induce a full complement of hair cell genes in supporting cells. A number of transcription factors have been proposed to co-operate with Atoh1 during hair cell induction, such as Gfi1, Pou4f3, Isl1, Gata3 and targets of the Wnt signaling pathway (*Costa and Henrique, 2015*; *Costa et al., 2015*; *Kuo et al., 2015*; *Walters et al., 2017*; *Yamashita et al., 2018*). Systematic combinatorial testing of these transcription factors in the context of a better defined hair cell gene regulatory network will be required to define the best cocktail of hair cell transcription factors required for optimum reprogramming of a hair cell fate.

# Materials and methods

## Animals

*Atoh1-CreERT2* (MGI: Tg(Atoh1-cre/Esr1*)14Fsh) (*Machold and Fishell, 2005*) and Ai3 (MGI: Gt (ROSA)26Sor$^{tm3(CAG-EYFP)Hze}$/J) (*Madisen et al., 2010*) transgenic lines were obtained from Jackson Laboratories (stock numbers 007684 and 007903). *Atoh1$^{A1GFP/A1GFP}$* (MGI: Atoh1$^{tm4.1Hzo}$; Jackson Laboratories stock number 013593) mice were generated as previously described (*Rose et al., 2009*) and obtained from Dr. Huda Zoghbi, Baylor College of Medicine. *Gfi1-Cre* (MGI: Gfi1$^{tm1(cre)}$ $_{Gan}$) mice were generated as previously described (*Yang et al., 2010*) and obtained from Dr. Lin Gan, University of Rochester. *Lfng-GFP* (MGI: Tg(Lfng-EGFP)HM340Gsat) were generated by the GENSAT project (*Geschwind, 2004*; *Heintz, 2004*; *Schmidt et al., 2013*). ICR mice were used for utricle viral infection and culture. Males and females were used between 4 weeks to 8 weeks of age unless stated otherwise. To label adult utricle hair cells, *Atoh1-CreERT2* or *Gfi1-Cre* males were crossed with Ai3 homozygous females. Tamoxifen was dissolved in peanut oil at a concentration of 10 mg/ml; three doses of 75–100 mg/kg were subcutaneously injected into postnatal pups at 10, 12, and 14 days of age. The Baylor College of Medicine Institutional Animal Care and Use Committee approved all animal experiments.

## Utricle culture and adenoviral infection

Adenovirus serotype 5 (Ad5) was engineered with an AdenoX vector (*Mizuguchi and Kay, 1998*) to drive tdTomato alone or with Atoh1 flanking with a T2A sequence under the control of the Ef1a promoter (pICPIS-EF1; Addgene catalog number 73355). Utricle explants were isolated from adult ICR mice as previously described (*Brandon et al., 2012*). After dissection, utricles were cultured on polycarbonate membrane (SPI supplies) in DMEM-F12 with N2 supplement, penicillin and Fungizone. All cultures were maintained in a 5% $CO_2$ humidified incubator. For gentamicin treatment, cultures were maintained in 0.1 μM, 0.5 uM, 1 μM or 2 μM of gentamicin medium for 24 hr. For virus infection, gentamicin-containing medium was replaced with 25 μl medium containing $5 \times 10^9$ viral particles one day later. Utricle explants were incubated in the viral solution for one hour at 37°C. One hour after viral incubation, 175 μl of serum-free medium was added to each well and virus-containing medium were replaced with DMEM-F12 with 5% FBS one day later. Utricles were cultured for additional 1–9 days and half the medium was replaced with fresh medium every other day. For all experiments, three biological replicates (i.e. parallel measurements of biologically distinct samples) were used.

## Detection of caspase activity

For detection of caspase activity, utricles were cultured in the presence or absence of 0.1 mM gentamicin for 24 hr. After removing the gentamicin-containing medium, 50 μl of NucView488 Caspase-3 substrate (*Cen et al., 2008*; Biotium Inc, USA) was added to the explants to give a final concentration of 5 μM for 30 min. The utricles were then washed three times with PBS for 10 min each at 37°C, fixed and processed for immunohistochemistry.

## Immunohistochemistry

For utricle staining, explants were fixed in 4% paraformaldehyde for one hour and washed with PBS containing 0.1% TritonX-100. Primary antibodies in this study used were anti-Moysin7a (1:500, rabbit; Proteus), anti-Sox2 (1:500, rabbit; Millipore), anti-RFP (1:500, Rabbit; Millipore), anti-GFP (1:500, chicken; Abcam) and anti-CD326 (1:200; rat; Thermo Fisher; Catalog number 17-5791-82). Secondary antibodies used were Alexa Fluor 488 and Alexa Fluor 594 (1:2000, Invitrogen). Cell nuclei were labeled by incubation in 0.5 ug/mL of DAPI in PBS for 20 min. Immunofluorescence images were captured on a Zeiss AxioImager microscope with Apotome structured illumination..

## Western blotting

Primary antibodies used in this study were anti-RFP (1:5,000, rabbit; Millipore), anti-Atoh1 (1:50,000, chicken; a gift from Matthew Kelley and Thomas Coate; *Driver et al., 2013*) and anti-GAPDH (1:5,000, chicken; Millipore). Experiments were performed under standard Western protocol with

Amersham ECL western blotting detection (GE Healthcare). Images were captured by a LAS-4000 Mini luminescent image analyzer.

## Purification of cells by FACS

Cochleas were dissected from P1 *Atoh1*<sup>A1GFP/A1GFP</sup> or P21 *Lfng-GFP* mice for collecting neonatal hair cells or adult supporting cells respectively. For collecting adult utricle hair cells, utricles were dissected from *Atoh1-CreERT2;* Ai3 mice after treated with tamoxifen as described above. For collecting adult utricle supporting cells, utricles were dissected from *Gfi1-CreER;* Ai3 mice. Isolated cochleae or utricles were then dissociated with the papain dissociation system (Worthington Biochemical Corp). Briefly, tissues were washed with calcium and magnesium-free PBS and then incubated in papain solution for 50 min at 37°C with 200 rpm shaking. The papain solution was removed and the tissue rinsed in CMF-PBS with 2% FBS. The tissue was then gently triturated 100–150 times with a 1000 µl pipette tip in CMF-PBS containing 2% FBS to generate a single cell suspension. For isolating adult utricle supporting cells, dissociated cells were stained with anti-CD326 conjugated with Allophycocyanin (APC) for 20 min in 4°C before sorting. Cells were purified on a BD FACS Aria cell-sorting flow cytometer using a 100 µm nozzle. For RNA-seq, approximately 2000 cells were sorted and collected directly into lysis buffer. For ATAC-seq, approximately 5000 cells were sorted and collected in DMEM + 10% FBS. For single cell RNA-seq, live cells were sorted on the basis of tdTomato fluorescence and individual cells were manually picked under a fluorescence microscope and palced in lysis buffer.

## QPCR

Total RNA was extracted from sorted cells using an Arcturus PicoPure RNA isolation kit (Applied Biosystems) following the manufacturer's instructions. cDNA was generated using qScript cDNA SuperMix (Quantabio). Quantitative PCR (qPCR) was performed with Master SYBR Green Kit (Applied Biosystems) on a Step One Plus real-time PCR system (Applied Biosystems). Relative quantification (2<sup>-ddCT</sup>) of gene expression was analyzed with the housekeeping gene GAPDH as an internal control. Gene specific primers using for qPCR were as following: tdTomato-F (CCT GTT CCT GGG GCA TGG) and tdTomato-R (TGA TGA CGG CCA TGT TGT TG); Atoh1-F(ATG CAC GGG CTG AAC CA) and Atoh1-R (TCG TTG TTG AAG GAC GGG ATA); Myosin6-F (TGT TAA GGC AGG TTC CTT GAA G) and Myosin6-R (ACA CCA GCT ACA ACT CGA AAC); Myosin7-F (AGG GGG ACT ATG TAT GGA TGG A) and Myosin7-R (ATG TGC GTG GCA TTC TGA GG); GAPDH-F (AGG TCG GTG TGA ACG GAT TTG) and GAPDH-R (TGT AGA CCA TGT AGT TGA GGT CA).

## RNA-seq

Approximately 2,000 FACS-sorted cells were used as input for RNA-seq. RNA extraction was performed using a SMART-seq v4 Ultra Low input RNA kit. The extracted and amplified cDNA was measured with Agilent Bioanalyzer and DNA libraries prepared with the Nextera XT DNA Library preparation kit. Paired-end (75 × 75 bp) sequencing was performed on an Illumina Nextseq500 instrument. The number of biological replicates used were: Utricle hair cells (2), Utricle supporting cells (3), tdTomato-infected utricles (3), Atoh1-tdTomato infected utricles (2), cochlear supporting cells (2). Each sample was sequenced to a depth of approximately 30 million reads. Sequencing data have been deposited in GEO under accession codes GSE122732 and GSE121610.

## RNA-seq analysis

RNA-seq datasets were uploaded to the Galaxy web platform for data analysis (*Afgan et al., 2016*). Briefly, processed paired-end reads were mapped to the mouse reference genome (mm10) using HISAT2. Read counts were calculated through HTseq-count and differentially expressed genes were identified using the DESeq2 package. Adjusted p values were calculated in the DESeq2 package using a Benjamini-Hochberg correction. Genes with fold change more than four folds and adjusted P-value less than $1 \times 10^{-5}$ were considered significant. Identification of transcripts enriched in utricle hair cells and supporting cells was calculated by direct comparison of RNA-seq reads for each cell type. Transcripts enriched in Ad-tdTomato infected cells were identified by comparing RNA-seq reads between utricle supporting cells and Ad-tdTomato cells. Similarly, transcripts enriched in Ad-tdTomato-Atoh1 were identified comparing RNA-seq reads between utricle supporting cells and

Ad-tdTomato-Atoh1 infected cells. For principle component analysis of RNA-seq data, normalized counts for each groups were used as input data in Clustvis. The input data were pre-processed with Pareto scaling and singular value decomposition with imputation. The DAVID web-tool (*Huang et al., 2009b*; *Huang et al., 2009a*) was used to identify enriched GO terms.

## Single cell MATQ-seq

Ad-tdTomato or Ad-tdTomato-Atoh1 infected supporting cells were FAC-sorted in culture medium. Single cells were individually picked into 96-well PCR plate (Bio-rad Cat. No. HSS9601) containing 1 µL of MATQ-seq lysis buffer (0.65 µL of 0.3% Triton-X100 ultrapure water (Thermo Fisher Scientific, Cat. No. 750024), 0.2 µL of MATQ-seq primer mix (*Sheng et al., 2017*), 0.05 µL of dNTP (NEB, Cat. No. N0447S), 0.1 M DTT 0.05 µL, and 0.05 µL RnaseOUT (Thermo Fisher Scientific, Cat. No. 10777019)) and 10 µL of PCR-grade mineral oil (Sigma Cat. No. M8662) to prevent evaporation. MATQ-seq was performed on a Bravo Automated Liquid Handling Platform (Agilent). The lysis plate was then briefly centrifuged. Reverse transcription was performed as previously described (*Sheng et al., 2017*). To distinguish the two strands, second strand synthesis was performed with a primer containing three mismatches to the first strand primer at the 3' end. The PCR products were then purified with 1.2x AMPure XP beads (Beckman Coulter, Cat. No. A63880).

To make double stranded cDNA, 20 ng of PCR product was diluted into 10µLof PCR grade water in each well of a 96-well plate and heated at 95℃ for 30 s to melt the DNA. To perform the reaction, 10 µL of enzyme mix containing 2 µL 10X Thermopol Buffer, 0.6 µL 10 µM barcoded primer, 0.5 µL 10 mM dNTPs, 0.3 µL Deepvent exo-DNA polymerase (NEB, Cat. No. M0259S), and 6.6 µL PCR grade water were added to each sample. The barcoded primer contains sequences specific to the first strand primer and a P7 sequence compatible with Illumina sequencer. 20 cycles of 20 s at 60℃ and 30 s at 72℃ were performed to add the barcoded primer to the first strand. 5 µL of 50 mM EDTA was then added to each well to terminate the reaction. Samples were then pooled together and purified with 1x AMPure XP beads. 50 ng of the purified product was then tagged at 55℃ for 5 min using the Nextera DNA library prep kit (Illumina, Cat. No. FC-121–1030). Libraries were then amplified with Nextera P5 primers and P7 primers supplied by the manufacturer to select only the amplicons with Unique Molecular Identifiers (UMI) on the first strand. A Duplex-specific Nuclease (Evrogen, Cat. No. EA003) treatment was performed on 100 ng of the library to remove the ribosomal cDNA as previously described (*Sheng et al., 2017*). The samples were sequenced on an Illumina Nextseq500. Sequencing data have been deposited in GEO under accession codes GSE127683.

## Single cell MATQ-seq analysis

For single cell data analysis, nine bases were trimmed from read two to remove amplicon index bases. Pair-end mapping was performed against the mouse genome (mm10) using STAR aligner. Gencode annotation release M10 (GRCm38.p4) was used for transcript annotation. Only reads with mapping quality over 250 were used for downstream analysis. Barcode retrieval, unique barcode counting, and gene expression quantification were performed as previously described (*Sheng et al., 2017*). The mapping position of the reads was included as part of the identity of the corresponding barcodes. Only reads mapped to the exon region with correct strandness were used for gene expression quantification. tSNE plots were generated using MATLAB. Heat maps were generated using the clustergram function in MATLAB.

## ATAC-seq

Approximately 5,000 FACS-sorted cells were used as input for ATAC. ATAC was performed according to *Corces et al. (2016)*. Briefly, sorted cells were spun down, FACS buffer was removed, the pellet was re-suspended in a transposase-containing reaction mixture complete with 0.05% digitonin prior to tagmentation at 37℃ with 1000 rpm agitation for 30 min. Next, transposed DNA was purified with a Qiagen PCR MinElute kit (Qiagen 28004). Fast-ATAC libraries were purified with a 1.8X SPR purification using AMPure XP (Beckman Coulter) beads following PCR amplification. Paired-end (75 × 75 bp) sequencing was performed on an Illumina Nextseq500 instrument. The number of biological replicates used were: Utricle hair cells (3), utricle supporting cells (3), tdTomato-infected utricles (3), Atoh1-tdTomato infected utricles (2), cochlear supporting cells (2). Each sample was

sequenced to a depth of approximately 30 million reads. Sequencing data have been deposited in GEO under accession codes GSE121610.

## ATAC analysis

ATAC-seq reads were mapped to the mouse genome (mm10) using Bowtie2 with default paired-end settings. All non-nuclear and unmapped paired reads were discarded. Duplicated reads were removed with the picard MarkDuplicates function, using default settings. Peak calling for analysis was carried out with Macs2 on the merged BAM file, Macs2 callpeak –nomodel –broad. Blacklisted regions identified by ENCODE from mm9 were lifted over to mm10 and then removed from the comprehensive peak file with the bedtools subtract module. Reads were counted for each condition from the comprehensive peak file using bedtools. Peak annotation was performed using annotate-Peaks.pl (Homer). The annotated compiled peak files were used to identify which genes containing accessible peaks. For peak accessibility comparison between each condition, we isolated the called peaks that are assigned to specific groups of genes. We designated all peaks greater than 2 kb away from an annotated TSS to be enhancers. We then identified the unique and overlapping peaks between each group using bedtools. Aggregation plots were performed using annotatePeak.pl (Homer). Two normalizations were performed on the samples before the analysis. The read length of the samples was adjusted to 40 bp and the profiles were then normalized based on a set of common highly accessible region (i.e. about 100 peaks from top 1000 peaks of each sample). PCA and differential accessibility analysis were performed with the DESeq2 R package using the multicov file as input. Conditional quantile normalization was performed with the cqn R package. Motif enrichment analysis and individual condition peak calling was conducted with Homer. ATAC-seq signal heat maps were generated in Homer and visualized using Java TreeView. Bed graph generation for UCSC browser visualization of Fast-ATAC signals was performed with Homer, and all reads were normalized by read count, where scores represent read count per bp per $1 \times 10^7$ reads.

## Enhancer motif analysis

MEME suite 5.0.1 was used for de novo motif discovery and motif analysis of Atoh1 targets. Enhancer regions were extracted from the UCSD browser as an input file. The Atoh1 binding motif was extracted from the JASPAR database as the motif input. FIMO (find individual motif occurrence) were used to identify the occurrence of Atoh1 motifs on our input sequences. Subsequently, MEME was used to discover reoccurrence motifs in candidate enhancer regions and Tomtom application was used to match identified motifs to known motifs.

## Acknowledgements

We thank Alyssa Crowder for expert technical assistance and past and present members of the Groves lab for help and advice, particularly Dr. Rende Gu for his invaluable assistance with demonstrating utricle dissection. We thank Huda Zoghbi and Lin Gan for providing mice, and for Noah Shroyer and Min-Shan Chen for making mice available to us at Baylor. We thank Matt Kelley and Tom Coate for their gift of Atoh1 antibodies, Jr-Han Tai for assistance with data analysis and Yung-Hsin Huang for assistance with sorting. We thank the Gene Vector Core at Baylor College of Medicine and the expert assistance of Dr. Kazuhiro Oka, and the Cytometry and Cell Sorting Core at Baylor College of Medicine and the expert assistance of Joel M Sederstrom, Amanda White and Bethany Tiner. At USC we thank Welly Makmura and the Broad Stem Cell Flow Cytometry Facility for expert technical assistance.

## Additional information

### Funding

| Funder | Grant reference number | Author |
| --- | --- | --- |
| National Heart, Lung, and Blood Institute | F31HL136065 | Matthew C Hill |
| National Institutes of Health | DP2EB020399 | Chenghang Zong |

| | | |
|---|---|---|
| Eunice Kennedy Shriver National Institute of Child Health and Human Development | RO1DE023177 | James F Martin |
| National Heart, Lung, and Blood Institute | RO1HL127717 | James F Martin |
| National Heart, Lung, and Blood Institute | RO1HL130804 | James F Martin |
| National Heart, Lung, and Blood Institute | RO1HL118761 | James F Martin |
| Vivian L Smith Foundation | | James F Martin |
| MacDonald Research Fund | 16RDM001 | James F Martin |
| Fondation Leducq | Transatlantic Network of Excellence Award (14CVD01) | James F Martin |
| National Institute on Deafness and Other Communication Disorders | RO1DC015829 | Neil Segil |
| Hearing Health Foundation | Hearing Restoration Project Consortium Grant | Neil Segil |
| National Cancer Institute | CA125123 | Andrew K Groves |
| National Institute on Deafness and Other Communication Disorders | RO1DC014832 | Andrew K Groves |

The funders had no role in study design, data collection and interpretation, or the decision to submit the work for publication.

## Author contributions

Hsin-I Jen, Conceptualization, Formal analysis, Validation, Investigation, Methodology, Writing—original draft, Writing—review and editing; Matthew C Hill, Kuanwei Sheng, Data curation, Formal analysis, Investigation, Methodology, Writing—review and editing; Litao Tao, Data curation, Formal analysis, Investigation, Methodology, Writing - review and editing; Wenjian Cao, Data curation, Formal analysis, Investigation, Methodology; Hongyuan Zhang, Validation, Investigation; Haoze V Yu, Formal analysis, Methodology; Juan Llamas, Investigation, Methodology; Chenghang Zong, Formal analysis, Funding acquisition, Writing—review and editing; James F Martin, Funding acquisition, Methodology, Writing—review and editing; Neil Segil, Conceptualization, Formal analysis, Funding acquisition, Writing—review and editing; Andrew K Groves, Conceptualization, Supervision, Funding acquisition, Methodology, Writing—original draft

## Author ORCIDs

Andrew K Groves http://orcid.org/0000-0002-0784-7998

## Ethics

Animal experimentation: This study was performed in strict accordance with the recommendations in the Guide for the Care and Use of Laboratory Animals of the National Institutes of Health. All of the animals were handled according to approved institutional animal care and use committee (IACUC) protocols AN-4956 (Baylor College of Medicine) and 20862 (University of Southern California).

## Decision letter and Author response

Decision letter https://doi.org/10.7554/eLife.44328.031
Author response https://doi.org/10.7554/eLife.44328.032

## Additional files

### Supplementary files
• Transparent reporting form
DOI: https://doi.org/10.7554/eLife.44328.025

### Data availability
Sequencing data have been deposited in GEO under accession codes GSE122732 and GSE121610.

The following datasets were generated:

| Author(s) | Year | Dataset title | Dataset URL | Database and Identifier |
|---|---|---|---|---|
| Jen H-I | 2018 | RNA-seq of adult utricle hair cells and supporting cells | https://www.ncbi.nlm.nih.gov/geo/query/acc.cgi?acc=GSE122732 | NCBI Gene Expression Omnibus, GSE122732 |
| Jen H-I, Groves AK | 2018 | ATAC-seq of adult utricle hair cells and supporting cells and P21 cochlear supporting cells | https://www.ncbi.nlm.nih.gov/geo/query/acc.cgi?acc=GSE121610 | NCBI Gene Expression Omnibus, GSE121610 |

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
