## [Decision Letter]

Thank you for submitting your article "Transcriptomic and epigenetic regulation of hair cell regeneration in the mouse utricle and its potentiation by Atoh1" for consideration by *eLife*. Your article has been reviewed by Didier Stainier as the Senior Editor, Francois Guillemot as the Reviewing Editor, and three reviewers.

The reviewers have discussed the reviews with each other and the Reviewing Editor has drafted this decision to help you prepare a revised submission.

Summary:

This study examines the capacity of transcription factor Atoh1 to reprogram cells in damaged mouse vestibules into hair cells. The authors use RNA-Seq and ATAC-Seq analyses to show that Atoh1 fails to induce the transcription of a substantial number of hair cell-specific genes, including genes with opened chromatin, suggesting that essential co-regulators are missing for efficient repair.

The consensus from the referees is that this is a timely, well-conceived and well-executed study and that its conclusions are mostly supported by the data. However, the reviewers have identified several issues which will need to be resolved prior to publication. This should require new analysis of the data and changes in the text but little or no new.

Essential revisions:

1) The authors use two different genetic labeling strategies to isolate hair and support cells from the utricle for differential expression analysis described in Figure 1. The purity of these cell populations is uncertain. What fraction of GFP+ cells isolated from the utricle of Atoh1CreER; Ai3 mice were contaminated with support cells, and what fraction of GFP- CD326+ cells were contaminated with hair cells? This information is necessary to rigorously evaluate the sensitivity and specificity of the approach.

2) The number of hair cells lost in utricle explants cultured in the presence and absence of 0.1 mM gentamicin for 24 hours should be quantified.

3) The total number of support cells transduced with Ad-Atoh1-tdTomato that express Myo7a should be reported, rather than just documenting the percent of double labeled cells as shown in Figure 2F.

4) It is surprising that the massive increase in Atoh1 protein levels observed in Ad-Atoh1 compared to Ad-Tom infected cells (Figure 2C) is not represented by a similar fold increase in Atoh1 mRNA (Figure 2I). What might account for this disparity?

5) The authors primarily focus on uHC genes that are activated in uSCs in response to damage and Atoh1 transduction (Figure 3). It would also be informative to characterize uSC gene sets that are, and are not, repressed by damage and Ad-Atoh1 transduction, as they may also represent important barriers to HC transdifferentiation. Some of this data is reported for the single cell analysis in Figure 4—figure supplement 2, but it may be worthwhile performing a more detailed characterization from the bulk RNA-seq data, highlighting candidate genes of interest.

6) Data in Figure 3B and C indicate that some genes involved in stereocilia bundle formation are upregulated in response to Ad-Atoh1 infection. Authors should comment on whether HC like-cells exhibit morphological features of uHCs and provide supporting evidence.

7) Do the 428 common hair cell genes mentioned in subsection “Differences in the regenerative ability of utricle and cochlear supporting cells correlate with chromatin accessibility at the promoters of hair cell genes” show near-zero expression in uSC and cSC? If not, the differential chromatin accessibility between uSC and cSC at promoters and distal elements of these genes may reflect differences in mRNA expression levels. A correlation analysis should be performed between ATAC-seq and RNA-seq data sets for each of the two cell types.

9) Is the open chromatin environment in uSCs specific for uHC genes?

10) Results in Figure 6 highlight examples where ectopic expression of Atoh1 in damaged uSCs induces chromatin accessibility at uHC specific sites that may or may not be associated with gene activation of the nearby gene. Does the activation of uHC specific genes by Atoh1 always correlate with gains in chromatin accessibility at endogenous sites? Cellular reprogramming by forced expression of transcription factors sometimes involves the activation of target genes through the use of cryptic transcription factor binding sites (Johnson et al., 2018). How often does Atoh1 activate HC specific genes through the use of ectopic sites of chromatin accessibility, i.e. ones that do not normally appear in uHCs? It may be of interest to report on these findings.

11) The Atoh1 dependent ATAC-seq peak in the terminal exon of Kcnj13 is intriguing (Figure 6C), especially given the role of exonic enhancers in the regulation of neighboring genes (Birnbaum et al., 2012). Have the authors explored the possibility that other genes in the vicinity of Kcnj13 are ectopically expressed in response to Atoh1 transduction? The authors suggest that additional transcription factors may be required to activate some uHC specific genes that are unresponsive to Atoh1, particularly in cases where ATAC-seq peaks are gained in the vicinity of uninduced genes (Figure 6—figure supplement 3). The authors should acknowledge the alternative possibility that the nearest HC specific gene may not always be the target of the putative Atoh1 dependent enhancer.

12) A previous study on hair cell differentiation in adult mouse utricles (Lin et al., 2011; referenced in this manuscript) showed that 18 days was required for development of a clear differentiated hair cell phenotype with stereocilia development. Also, several studies in recent years have suggested that transient rather than constitutive overexpression of Atoh1 might be necessary for full differentiation of hair cells. These previous studies should be acknowledged.

13) Was the selection of 0.1 mM vs 1 mM gentamicin based on "abnormal, elongated morphology" of supporting cells after 24hour treatment with 1 mM gentamicin? Other than the image provided Figure 2—figure supplement 1B, how were differences between the supporting cells quantified? Since 1 mM leads to significantly more hair cells loss than 0.1 mM and larger areas in the sensory epithelium to be 'filled' by the remaining supporting cells, isn't this expected? Without quantifying a statistically significant difference between the two conditions the current reasoning is not justified.

---

## [Author Response]

Essential revisions:1) The authors use two different genetic labeling strategies to isolate hair and support cells from the utricle for differential expression analysis described in Figure 1. The purity of these cell populations is uncertain. What fraction of GFP+ cells isolated from the utricle of Atoh1CreER; Ai3 mice were contaminated with support cells, and what fraction of GFP- CD326+ cells were contaminated with hair cells? This information is necessary to rigorously evaluate the sensitivity and specificity of the approach.

To address this question, we sorted hair cells or supporting cells from the utricle using our two sorting protocols and then re-analyzed the sorted cells:

- For isolating hair cells using Atoh1CreER; Ai3 mice, we stained the utricle cells using APC conjugated CD326 to mark epithelial cells; here, all the APC+ve, GFP–ve cells would be considered contaminated supporting cells. We observe less than 1% of contaminating supporting cells.

- For supporting cells isolated from Gfi1-Cre; Ai3 mice and counterstained with CD326, we re-sorted the GFP -ve / CD326+ve population and analyzed GFP expression. Once again, less than 1% of the sorted cells expressed GFP, suggesting minimal hair cell contamination. We have now included the results of the re-sorting as a new Figure 1—figure supplement 2. We can include this as a figure supplement if the reviewers wish.

The RNA-seq data obtained from our sorting was used to generate lists of utricle hair cell- or supporting cellenriched genes. When we compared the RNA-seq reads between our purified hair cells and supporting cells, we found many genes that differ by more than 5000 fold between these two populations. This observation, together with our re-sorting data above suggests that the lists of hair cell and supporting cell-specific genes are likely to be robust.

2) The number of hair cells lost in utricle explants cultured in the presence and absence of 0.1 mM gentamicin for 24 hours should be quantified.

We cannot live label and follow hair cells in our explants, and so we are unable to quantify the numbers of hair cells lost from each explant during the first 24 hours of culture with gentamicin. However, we have now measured cell death and hair cell loss compared to untreated controls in the 0-24hour time period using caspase3 substrates previously used by Cunningham and colleagues to monitor cell death in the utricle. This data is now included as a new supplementary figure, Figure 2—figure supplement 2. We note that cell death continues in the cultures after the first 24 hours, but the new data gives a reasonable impression of the effects of this moderate dose of gentamicin in the first 24 hours of treatment.

3) The total number of support cells transduced with Ad-Atoh1-tdTomato that express Myo7a should be reported, rather than just documenting the percent of double labeled cells as shown in Figure 2F.

The number of supporting cells transduced with virus varies in each explant, so we have included data in the Results section showing the range of TdTom+/Myo7a+ cell numbers in our cultures as a new panel in Figure 2, together with the mean values. We feel that giving both the range and mean will give the reader the best impression of how our infection and reprogramming rates varied between explants.

4) It is surprising that the massive increase in Atoh1 protein levels observed in Ad-Atoh1 compared to Ad-Tom infected cells (Figure 2C) is not represented by a similar fold increase in Atoh1 mRNA (Figure 2I). What might account for this disparity?

The protein data in Figure 2C was obtained by infecting 293T cells and analyzing 2 days later, whereas the QPCR data in Figure 2I was obtained from infected utricles analyzed after 10 days, and so the two panels are not directly comparable. We were insufficiently clear in describing this, but we have now revised both the text and legend for Figure 2 to hopefully explain this better.

5) The authors primarily focus on uHC genes that are activated in uSCs in response to damage and Atoh1 transduction (Figure 3). It would also be informative to characterize uSC gene sets that are, and are not, repressed by damage and Ad-Atoh1 transduction, as they may also represent important barriers to HC transdifferentiation. Some of this data is reported for the single cell analysis in Figure 4—figure supplement 2, but it may be worthwhile performing a more detailed characterization from the bulk RNA-seq data, highlighting candidate genes of interest.

We have now performed this analysis using the same approach shown in Figure 3. This new data is shown as Figure 3—figure supplement 2. We find a similar trend to that reported in Figure 3B – we find that culturing damaged utricles alone down-regulates a significant number of supporting cell genes, and that Atoh1 further down-regulates an additional, small number of supporting cell genes. Supporting cell genes are generally much less well characterized than hair cell genes, and so none of the down-regulated genes jumped out as being noteworthy based on previously published studies. As the reviewer suggests, a significant number of our ~2600 supporting cell genes are NOT down-regulated by either damage and culture or Atoh1 transduction. It is indeed interesting to speculate whether expression of these recalcitrant supporting cell genes may serve as a roadblock to hair cell reprogramming, and whether additional hair cell transcription factors (such as Gfi1 and Pou4f3) may help overcome these impediments. We have expanded on this point in the Discussion section.

6) Data in Figure 3B and C indicate that some genes involved in stereocilia bundle formation are upregulated in response to Ad-Atoh1 infection. Authors should comment on whether HC like-cells exhibit morphological features of uHCs and provide supporting evidence.

We did not observe any obvious morphological specializations of the trans-differentiating supporting cells by light microscopy (for example, with fluorescently labeled phalloidin), either with or without Atoh1. This is consistent with the recent study from the Forge lab in which human utricles were transduced with Atoh1 adenovirus. In that study, the authors used SEM and TEM to examine their cultures, but were only able to observe small apical microvillar projections, together with occasional small projections reminiscent of kinocilia. It is possible that the transduced cells in both studies might develop more hair cell-like features with time (for example, the Lin et al., 2011 paper referenced below). Our lab is moving away from cultured utricle experiments to in vivo models of transcription factor reprogramming using transgenic mice carrying Creinducible transcription factors (including Atoh1), but these studies are still at a very preliminary stage.

7) Do the 428 common hair cell genes mentioned in subsection “Differences in the regenerative ability of utricle and cochlear supporting cells correlate with chromatin accessibility at the promoters of hair cell genes” show near-zero expression in uSC and cSC? If not, the differential chromatin accessibility between uSC and cSC at promoters and distal elements of these genes may reflect differences in mRNA expression levels. A correlation analysis should be performed between ATAC-seq and RNA-seq data sets for each of the two cell types.

In general, the RPKM values in supporting cells are 1-2 orders of magnitude lower than hair cells, although it is always a matter of judgment whether to filter/label these values as “near zero”. To help the reader get an impression of these differences, we have provided examples of RPKM counts of 12 hair cell genes in hair cells and utricle and cochlear supporting cells in the main text and in a new graph (Figure 5—figure supplement 2). We also performed a global correlation analysis of RNA-seq and ATAC-seq data for each hair cell gene promoter. That suggests a very low correlation between chromatin accessibility of hair cell genes (ATAC-seq) and their RPKM in either cochlear or utricle supporting cells (r values of 0.02 and 0.03 respectively). That said, it is true that hair cell genes with higher expression levels in supporting cells display more variable chromatin accessibility signatures on average than their lowly expressed counterparts. This analysis is now provided as Figure 5—figure supplement 2.

9) Is the open chromatin environment in uSCs specific for uHC genes?

It is hard to provide a clear answer to this question, as open chromatin is associated with actively transcribed genes as well as genes that are accessible but not transcribed. For example, utricle supporting cell-specific genes will obviously be accessible in supporting cells, and supporting cells – like all cells – express a variety of housekeeping genes that are maintained in a transcriptionally accessible state. To try and address the reviewer’s question with respect to hair cell genes, we filtered out genes that were did not show significant differences in expression between hair cells and supporting cells. For the remaining genes showing differential expression, we determined how many ATAC-seq peaks could be assigned to hair cell loci versus other loci. From our uSC ATAC-seq data, we found 3018 accessible peaks that can be assigned to uHC loci, and 6500 accessible peaks assigned to uSC loci. As expected, we found 26114 accessible peaks that could be assigned to other genes, such as housekeeping genes. Therefore, we do not think uHC loci have a uniquely open chromatin environment in uSCs compared to other regions of the genome. We have not included this analysis in the paper, but can add it if the referee(s) wish.

10) Results in Figure 6 highlight examples where ectopic expression of Atoh1 in damaged uSCs induces chromatin accessibility at uHC specific sites that may or may not be associated with gene activation of the nearby gene. Does the activation of uHC specific genes by Atoh1 always correlate with gains in chromatin accessibility at endogenous sites? Cellular reprogramming by forced expression of transcription factors sometimes involves the activation of target genes through the use of cryptic transcription factor binding sites (Johnson et al., 2018). How often does Atoh1 activate HC specific genes through the use of ectopic sites of chromatin accessibility, i.e. ones that do not normally appear in uHCs? It may be of interest to report on these findings.

To answer this question we isolated peaks from utricle HCs and Atoh1 transduced supporting cells (Ad-Atoh1), which could be annotated to genes that we found to be activated by Atoh1 expression. Next, we filtered these peaks for those which contained Atoh1 target motifs. Overall, we identified 241 peaks containing Atoh1 sites in uHCs and 281 peaks containing Atoh1 sites in Ad-Atoh1 supporting cells (now shown as Figure 6—figure supplement 4A). Interestingly, only 70 peaks out of 542 (15.5%) overlapped between the two conditions. Next, we looked at the ATAC-seq signal from these 542 peaks and found that Ad-Atoh1 infected supporting cells do indeed possess high accessibility of sites not found in uHCs (Figure 6—figure supplement 3B). Thus, this analysis suggests that, as the reviewer suggests, Atoh1 reprogramming frequently activates HC specific genes through ectopic sites not observed in hair cells in vivo. We have now added this to the Results section and Discussion section.

11) The Atoh1 dependent ATAC-seq peak in the terminal exon of Kcnj13 is intriguing (Figure 6C), especially given the role of exonic enhancers in the regulation of neighboring genes (Birnbaum et al., 2012). Have the authors explored the possibility that other genes in the vicinity of Kcnj13 are ectopically expressed in response to Atoh1 transduction? The authors suggest that additional transcription factors may be required to activate some uHC specific genes that are unresponsive to Atoh1, particularly in cases where ATAC-seq peaks are gained in the vicinity of uninduced genes (Figure 6—figure supplement 3). The authors should acknowledge the alternative possibility that the nearest HC specific gene may not always be the target of the putative Atoh1 dependent enhancer.

This is an excellent point, and we have now emphasized this in the discussion. To address this, we also searched +/- 200kb around the Kcnj13 locus. which includes Gigyf2, Efhd1 and Ngef. However, none of these genes show differences in expression between Ad-tdTomato infected cells and Ad-Atoh1 infected cells. We also mention this in the Results section.

12) A previous study on hair cell differentiation in adult mouse utricles (Lin et al., 2011; referenced in this manuscript) showed that 18 days was required for development of a clear differentiated hair cell phenotype with stereocilia development. Also, several studies in recent years have suggested that transient rather than constitutive overexpression of Atoh1 might be necessary for full differentiation of hair cells. These previous studies should be acknowledged.

We have tried to more clearly emphasize the issues with down-regulation of Atoh1 versus constitutive overexpression in the Discussion section. Given the perdurance of doxycycline in the inner ear reported by Zuo and colleagues, it is likely that current tet-off technology may not be suitable for providing this temporally regulated expression of Atoh1 in the mature ear.

13) Was the selection of 0.1 mM vs 1 mM gentamicin based on "abnormal, elongated morphology" of supporting cells after 24hour treatment with 1 mM gentamicin? Other than the image provided Figure 2—figure supplement 1B, how were differences between the supporting cells quantified? Since 1 mM leads to significantly more hair cells loss than 0.1 mM and larger areas in the sensory epithelium to be 'filled' by the remaining supporting cells, isn't this expected? Without quantifying a statistically significant difference between the two conditions the current reasoning is not justified.

We were concerned that the abnormal morphology we saw in 1mM gentamicin might be reflected in abnormal patterns of gene expression in supporting cells. We chose not to pursue this rigorously with additional RNAseq experiments, simply because the 0.1mM treatment gave good hair cell death and supporting cell morphology looked similar to in vivo models of hair cell damage, such as the recent Bucks et al. paper from the Stone lab. In retrospect, we feel justified in using this concentration given that over 75% of utricle supporting cell genes do not change significantly after 10 days of this treatment, as we now show in Figure 3—figure supplement 2A.